# FOOLING ADVERSARIAL TRAINING WITH INDUCING NOISE

## ABSTRACT

Adversarial training is widely believed to be a reliable approach to improve model robustness against adversarial attack. However, in this paper, we show that when trained on one type of poisoned data, adversarial training can also be fooled to have catastrophic behavior, e.g., $< 1\%$ robust test accuracy with $> 90\%$ robust training accuracy on CIFAR-10 dataset. Previously, there are other types of noise poisoned in the training data that have successfully fooled standard training ($15.8\%$ standard test accuracy with $99.9\%$ standard training accuracy on CIFAR-10 dataset), but their poisonings can be easily removed when adopting adversarial training. Therefore, we aim to design a new type of inducing noise, named ADVIN, which is an irremovable poisoning of training data. ADVIN can not only degrade the robustness of adversarial training by a large margin, for example, from $51.7\%$ to $0.57\%$ on CIFAR-10 dataset, but also be effective for fooling standard training ($13.1\%$ standard test accuracy with $100\%$ standard training accuracy). Additionally, ADVIN can be applied to preventing personal data (like selfies) from being exploited without authorization under whether standard or adversarial training.

## 1 INTRODUCTION

In recent years, deep learning has achieved great success, while the existence of adversarial examples (Szegedy et al., 2014) alerts us that existing deep neural networks are very vulnerable to adversarial attack. Crafted by adding imperceptible perturbations to the input images, adversarial examples can dramatically degrade the performance of accurate deep models, raising huge concerns in both the academy and the industry (Chakraborty et al., 2018; Ma et al., 2020).

*Adversarial Training* (AT) is currently the most effective approach against adversarial examples (Madry et al., 2017; Athalye et al., 2018). In practice, adversarially trained models have been shown good robustness under various attack, and the recent state-of-the-art defense algorithms (Zhang et al., 2019; Wang et al., 2020) are all variants of adversarial training. Therefore, it is widely believed that we have already found the cure to adversarial attack, i.e., adversarial training, based on which we can build trustworthy models to a certain degree.

In this paper, we challenge this common belief by showing that AT could be ineffective when injecting some small and specific poisonings into the training data, which leads to a catastrophic drop for AT on CIFAR-10 dataset in the test accuracy (from 85% to 56% on clean data) and the test robustness (from 51% to 0.6% on adversarial data). Previously, Huang et al. (2021) and Fowl et al. (2021b) have shown that injecting some special noise into the training data can make *Standard Training* (ST) ineffective. However, these kinds of noise can be easily removed by AT, i.e., AT is still effective. While in this work, we are the first to explore whether there exists a kind of special and irremovable poisoning of training data that could make AT ineffective.

Specifically, we first dissect the failure of Huang et al. (2021) and Fowl et al. (2021b) on fooling AT and find that they craft poisons on a standardly trained model. As pointed out by Ilyas et al. (2019), ST can only extract non-robust features, which will be discarded in AT because it only extracts robust features. In view of this, we should craft poisons with robust features extracted from adversarially trained models, which may be more resistant to AT. However, only using robust features is not sufficient to break down AT because we find that AT itself still works well when taking robust-feature perturbations during training. The key point is that we need to utilize a *consistent* misclassified target label for each class, and only with this *consistent bias* can we induce AT to the desired misclassification. Based on this, we instantiate a kind of irremovable poisoning, ADVersarially Inducing Noise (ADVIN), for the training-time data fooling. ADVIN can not only degrade

standard training like previous methods but also successfully break down adversarial training for the first time. To summarize, our main contributions are:

- We are the first to study how to make adversarial training ineffective by injecting irremovable poisoning. It is more challenging since all previous fooling methods designed for standard training fail to work under adversarial training.
- We instantiate a kind of irremovable noise, called ADVersarially Inducing Noise (ADVIN), to poison data. Extensive experiments show that ADVIN can successfully make adversarial training ineffective and outperform ST-oriented methods by a large margin.
- We apply ADVIN to prevent unauthorized exploitation of personal data, where ADVIN is shown to be effective against both standard and adversarial training, making our privacy-preserved data *truly unlearnable*.

## 2 RELATED WORK

**Data poisoning.** Data poisoning aims at fooling the model to have a poor performance on clean test data by manipulating the training data. For example, Biggio et al. (2012) aims at poisoning an SVM model. While previous works mainly focus on poisoning the most influential examples using adversarial noise (Koh & Liang, 2017; Muñoz-González et al., 2017), these methods can only play a limited role in the destruction of the training process of DNNs. Recently, Huang et al. (2021) and Fowl et al. (2021b) propose error-minimizing noise and adversarial example noise, respectively, which lead standardly trained DNNs on them to have a test accuracy close to or even lower than random prediction. Unfortunately, their poisons can be removed by adversarial training. Therefore, we focus on how to generate poisons that could not be removed by adversarial training and deconstruct the training process at the same time, i.e., making adversarial training ineffective. Addition discussion about recentlt related work could be found in Appendix E

**Adversarial Attack.** Szegedy et al. (2014) has demonstrated the vulnerability of deep neural networks, which could be easily distorted by imperceptible perturbations. Typically, adversarial attacks utilize the error-maximizing noise (untargeted attack) to fool the models at test time (Goodfellow et al., 2015). Specifically, the adversarial examples can be divided into two categories, untargeted (Goodfellow et al., 2015; Madry et al., 2017) and targeted attack. Compared to the untargeted manner, targeted attack generates adversarial examples such that they are misclassified to the target class (different from the original label). While iterative untargeted attack (Madry et al., 2017) is more popular in solving the inner loop of adversarial training, some recent works find that targeted attack can achieve comparable, and sometimes better, performance (Xie & Yuille, 2020; Kurakin et al., 2017; Wang & Zhang, 2019).

## 3 THE DIFFICULTY ON FOOLING ADVERSARIAL TRAINING

Considering a $K$-class image classification task, we denote the natural data as $\mathcal{D}_c = \{(\boldsymbol{x}_i, y_i)\}$, where $\boldsymbol{x}_i \in \mathbb{R}^d$ is a $d$-dimensional input, and $y_i \in \{1, 2, \ldots, K\}$ is the corresponding class label. To learn a classifier $f$ with parameters $\boldsymbol{\theta}_t$, *Standard Training* (ST) minimizes the following objective on clean data, where $\ell_{CE}(\cdot, \cdot)$ denotes the cross entropy loss:

$$\min_{\boldsymbol{\theta}_t} L_{\mathrm{ST}}(\mathcal{D}_c, \boldsymbol{\theta}) = \min_{\boldsymbol{\theta}_t} \mathbb{E}_{(x_i, y_i) \sim \mathcal{D}_c} \ell_{\mathrm{CE}}(f_{\boldsymbol{\theta}_t}(\boldsymbol{x}_i), y_i). \tag{1}$$

Instead, *Adversarial Training* (AT) aims to improve robustness against adversarial attack by training on adversarially perturbed data, resulting in the following minimax objective,

$$\min_{\boldsymbol{\theta}_t} L_{\mathrm{AT}}(\mathcal{D}_c, \boldsymbol{\theta}) = \min_{\boldsymbol{\theta}_t} \mathbb{E}_{(x_i, y_i) \sim \mathcal{D}_c} \max_{\|\boldsymbol{\delta}^t\|_p \leq \varepsilon_t} \ell_{\mathrm{CE}}\left(f_{\boldsymbol{\theta}_t}(\boldsymbol{x}_i + \boldsymbol{\delta}^t), y_i\right), \tag{2}$$

where the sample-wise perturbation $\boldsymbol{\delta}^t$ is constrained in a $\ell_p$-norm ball with radius $\varepsilon_t$ and the inner maximization is typically solved by PGD (Madry et al., 2017).

**Fooling Standard Training (FST).** Intuitively, the goal of the poisoned data $\mathcal{D}_p$ is to induce standard training to learn a model on $\mathcal{D}_p$ with parameters $\boldsymbol{\theta}_t$ that is ineffective for classifying natural images from $\mathcal{D}_c$. However, their fooling can only work for standard training while being easily alleviated under adversarial training. In other words, their "unlearnable examples" are actually *learnable*. Specifically, Huang et al. (2021) adopt the *error-minimizing noise* generated with the following min-min optimization problem for fooling standard training:

$$\min_{\boldsymbol{\theta}_s} \mathbb{E}_{(\boldsymbol{x}_i, y_i) \sim \mathcal{D}_c} \min_{\|\boldsymbol{\delta}^p\| \leq \varepsilon_p} \ell_{\mathrm{CE}}\left(f_{\boldsymbol{\theta}_s}(\boldsymbol{x}_i + \boldsymbol{\delta}^p), y_i\right), \tag{3}$$

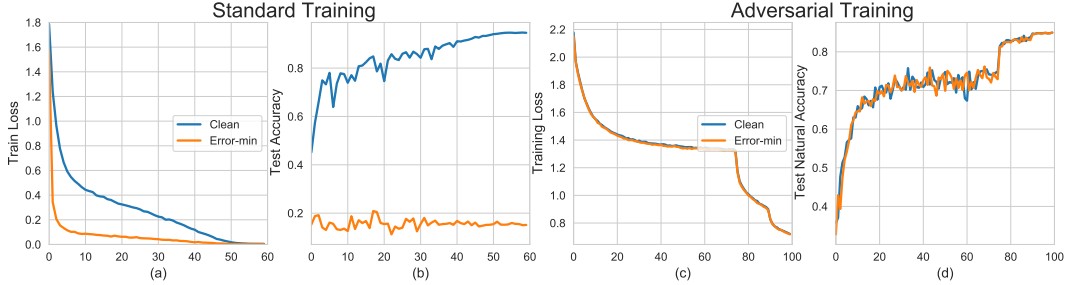

Figure 1: The training loss and natural test accuracy of models with 1) standard training on clean data (a) and error-minimizing poisoned data (b); 2) adversarial training on clean data (c) and error-minimizing poisoned data (d). All experiments are conducted with ResNet-18 on CIFAR-10 dataset.

where $\boldsymbol{\theta}_s$ is the *source* model that is used to generate poisons with perturbation radius $\varepsilon_p$. The inner loop seeks the $L_p$-norm bounded noise $\boldsymbol{\delta}$ by minimizing the loss with PGD steps, and the outer loop further optimizes the parameters $\boldsymbol{\theta}$ by minimizing the loss on the adversarial pair $(\boldsymbol{x}_i + \boldsymbol{\delta}^p, y_i)$.

To investigate how error-minimizing noise can fool standard training, we compare the training process of clean data and error-minimizing perturbed data in Figure 1(a)(b), where the training loss of error-minimizing data is significantly smaller. This indicates that error minimization is designed to minimize the loss of the perturbed pair $(\boldsymbol{x}_i + \boldsymbol{\delta}_i^p, y_i)$ to near zero such that the poisoned sample can not be used for model updating. While for adversarial training, as shown in Figure 1(c)(d), its inner maximization process can easily remove the error-minimizing noise by further lifting the loss of the perturbed pair $(\boldsymbol{x}_i + \boldsymbol{\delta}_i^p + \boldsymbol{\delta}_i^t, y_i)$ with the error-maximizing noise $\boldsymbol{\delta}_i^t$. In this way, the hidden information is uncovered and makes those unlearnable examples learnable again.

Thus, to fool adversarial training, we need to go beyond the paradigm of unlearnable examples and design a stronger type of poisoning, for which we need it to be *irremovable and resistant to error-maximizing perturbations*. Below, we introduce our attempts to design this irremovable noise.

## 4 DESIGNING OF IRREMOVABLE NOISE

Based on the investigation in Section 3, we can easily see that it is more challenging for fooling adversarial training than standard training. In the following, we will design effective irremovable noise from the aspects of features, labels, and training strategies.

### 4.1 THE NECESSITY OF ROBUST FEATURES

First, we notice that it is necessary to use robust features for fooling AT. Specifically, we compare poisons generated using Fowl et al. (2021b) from two different pre-trained models, a standardly trained model and an adversarially trained model, both with $\varepsilon_p = 32/255$. Note that although here we use a larger perturbation radius $\varepsilon_p$, this factor can only slightly fool AT by $\sim 10\%$ performance drop in Huang et al. (2021) and cannot guarantee irremovability. We compare the poisons generated from robust and non-robust features. The results are shown in Figure 2a. We can see that even with a larger $\varepsilon_p$, poisons generated from the ST source model are almost useless (orange lines). In contrast, the poisons generated from the robust source model can effectively bring down the final robustness from $\sim 50\%$ to $\sim 30\%$ (blue lines). More details of experiments for poison generations and training process could be found in Appendix A.2

This observation indicates that robust features are necessary for fooling AT. According to Ilyas et al. (2019), ST can only extract non-robust features, and thus the generated poisons only contain non-robust features, which, however, will be discarded under AT since it only relies on robust features. Therefore, to fool AT effectively, the source model itself must contain robust features so that the generated poisons could contain robust features that are resistant to AT. To achieve this goal, we adopt the adversarially trained models to craft poisons.

### 4.2 THE NECESSITY OF CONSISTENT LABEL BIAS

As shown in Section 3, the error-minimizing noise can be easily removed by the error-maximizing process of AT. Recalling that AT itself can learn good models with error-maximizing noise generated

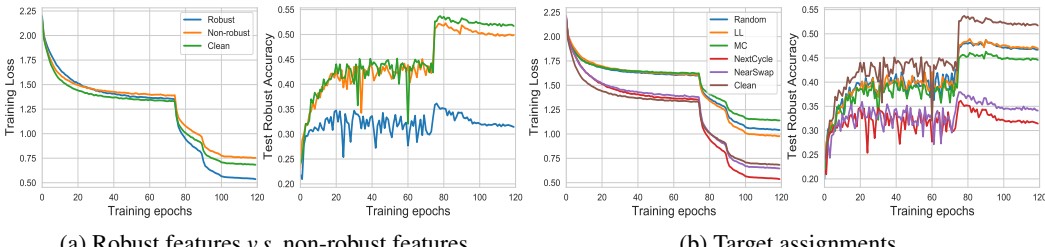

(a) Robust features *v.s.* non-robust features        (b) Target assignments

Figure 2: The training loss and robust test accuracy of adversarial training with poisoned data generated with (a) robust (AT) and non-robust (ST) pre-trained models; and (b) different target assignments. All experiments are conducted with ResNet-18 on CIFAR-10 dataset.

by itself using untargeted attack, we consider to use alternative target labels that are different from the error-maximizing objective. Formally, given a source model $f_{\boldsymbol{\theta}_s}$ and a natural pair $(\boldsymbol{x}_i, y_i) \in \mathcal{D}_c$, we pick a target class $y_i'$ and generate the poison $\boldsymbol{\delta}_i^p$ by

$$\boldsymbol{\delta}_i^p = \underset{\|\boldsymbol{\delta}_i^p\| \leq \varepsilon_p}{\arg\min} \ell_{\mathrm{CE}}\left(f_{\boldsymbol{\theta}_s}(\boldsymbol{x}_i + \boldsymbol{\delta}_i^p), y_i'\right). \tag{4}$$

Specifically, we consider the following strategies for assigning fooling labels:

- **Random** (Xie & Yuille, 2020): a randomly drawn label $y_i' \overset{u.a.r.}{\sim} \{1, 2, \ldots, K\}$;
- **LL** (Kurakin et al., 2017): the Least Likely label $y_i' = \arg\max_{y \neq y_i} \ell_{\mathrm{CE}}\left(f_{\boldsymbol{\theta}_s}(\boldsymbol{x}_i), y\right)$;
- **MC** (Wang & Zhang, 2019): the Most Confusing label $y_i' = \arg\min_{y \neq y_i} \ell_{\mathrm{CE}}\left(f_{\boldsymbol{\theta}_s}(\boldsymbol{x}_i), y\right)$;
- **NextCycle** (ours): the next label in a cyclic order $y_i' = (y_i + 1 \mod K)$;
- **NearSwap** (ours): label swapping with $y_i' = \begin{cases} y_i + 1 \mod K, & \text{if } y_i = 2k + 1, \\ y_i - 1 \mod K, & \text{if } y_i = 2k, \end{cases} \quad k \in \mathbb{N}.$

We list their performance against AT in Figure 2b. We can see that like error-minimizing and error-maximizing noise, both Random, LL, and MC methods also fail to poison AT (blue, orange, and green lines). Instead, we can see that both NextCycle and NearSwap can effectively degrade robust accuracy to $30\% - 35\%$ (red and purple lines). Comparing the five strategies, we can find a common and underlying rule for the effective ones, *e.g.,* NextCycle and NearSwap, that the label mapping $g : y_i \rightarrow y_i'$ is consistent among samples in the same class while being different for samples from different classes. As a result, they impose a *consistent bias* on the poisoned data such that all samples in the class A are induced to a specific class B. In this way, they can induce AT to learn a false mapping between features and labels, resulting in a low robust accuracy on test data. Details of noise generation for these five label mapping strategy can be seen in Appendix A.3.

### 4.3 TRAINING STRATEGY: INDUCING ADVERSARIAL TRAINING (IAT)

From the above two sections, we have known that poisons generated from robust models with consistent label bias can successfully fool AT to some extent. Nevertheless, we still notice there are some discrepancies between the fooling process and the adversarial training process. Specifically, for fooling, we utilize a pre-trained source model $f_{\boldsymbol{\theta}_s}$; while for AT, we train a target model $f_{\boldsymbol{\theta}_t}$ from scratch. Even though the source model is robust enough, its loss landscape could be very different from that of a randomly initialized target model. Besides, the source model is learned with clean data, while the target model is learned with poisoned data instead. These discrepancies between the source model $f_{\boldsymbol{\theta}_s}$ and the target model $f_{\boldsymbol{\theta}_t}$ will make the poisons generated from the source model less effective for fooling the target model.

To close these *source-target discrepancies*, we believe that it is better to generate poisons also from a randomly initialized model that is adversarially trained for predicting the target labels $y'$. This will lead to an alternating procedure between two steps:

a) generating poisons $\mathcal{D}_p$ from the source model $f_{\boldsymbol{\theta}_s}$ by

$$\boldsymbol{\delta}_i^p = \underset{\|\boldsymbol{\delta}_i^p\| \leq \varepsilon_p}{\arg\min} \ell_{\mathrm{CE}}\left(f_{\boldsymbol{\theta}_s}(\boldsymbol{x}_i + \boldsymbol{\delta}_i^p), y_i'\right), \forall \, x_i \in \mathcal{D}_c. \tag{5}$$

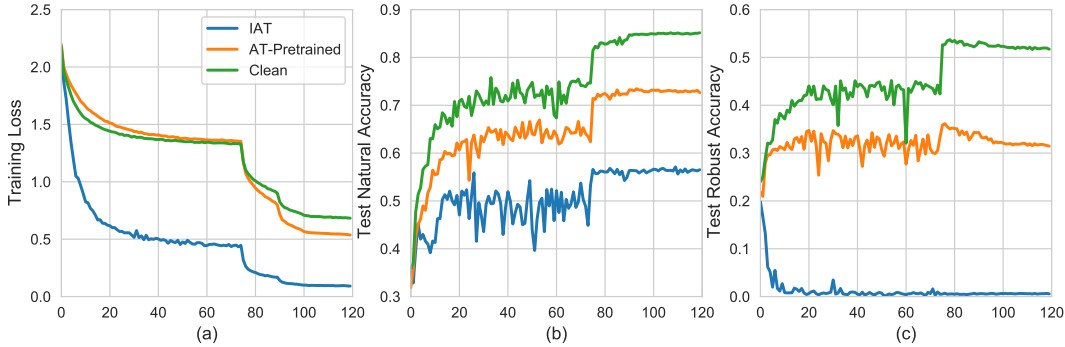

Figure 3: The training loss (a), the natural test accuracy (b), and the robust test accuracy (c) of adversarial training under clean datasets, AT-pre-trained poisons, and IAT poisons, respectively.

b) adversarial training of the source model $f_{\boldsymbol{\theta}_s}$ such that it could robustly predict the poisoned images $(\boldsymbol{x}_i + \delta_i^p)$ to the inducing target labels $y'$, *i.e.*,

$$\min_{\boldsymbol{\theta}_s} \mathbb{E}_{(\boldsymbol{x}_i + \boldsymbol{\delta}_i^p, y_i) \sim \mathcal{D}_p} \max_{\boldsymbol{\delta}} \ell_{\text{CE}} \left( f_{\boldsymbol{\theta}_s}(\boldsymbol{x}_i + \boldsymbol{\delta}_i^p + \boldsymbol{\delta}), y_i' \right). \tag{6}$$

In practice, we will keep involving the loop until the following Poisoning Success Rate (PSR)

$$\text{PSR}(\mathcal{D}_p) = \mathbb{E}_{(\boldsymbol{x}_i + \boldsymbol{\delta}_i^p, y_i) \sim \mathcal{D}_p} \mathbb{I} \left[ y_i' = \arg\max \ f_{\boldsymbol{\theta}_s}(\boldsymbol{x}_i + \boldsymbol{\delta}_i^p) \right] \tag{7}$$

reaches a certain threshold $\eta$. In this way, the source model will robustly classify the poisoned data to the target classes and generate poisons with desired inducing features. Therefore, we name this iterative poisoning process as Inducing Adversarial Training (IAT) as it involves the induction into the adversarial training process. As shown in Figure 3, when compared to poisoning with pre-trained models (blue line), our IAT (orange line) can achieve an even smaller training loss (plot a), while having worse natural test accuracy (plot b) and much worse robust test accuracy $31.4\% \rightarrow 0.6\%$ (plot c). This shows that our IAT is much better at fooling AT by causing worse test robustness while inducing a smaller training loss.

At last, combining IAT with our inducing label assignments, we arrive at our instantiation of irremovable noise, namely Adversarially Inducing Noise (ADVIN), that could induce AT to a catastrophic behavior. In particular, ADVIN could combine the advantages of using robust features and consistent label bias for fooling adversarial training. As shown in Eq. 5, the noise is generated by the source model through a targeted PDG attack towards the targeted label $y'$ with a consistent bias. Meanwhile, the source model is adversarially trained as shown in Eq. 6. In this way, we can inject the robust features of the targeted classes $y'$ into $y$-class samples consistently. When the poisoning success rate (PSR) reaches the given threshold, the poisons are trained to have enough robust features about the target class $y'$. Therefore, when the target model is trained on the $y$-class poisoned data, it will be induced to use the $y'$-class robust features in the perturbation to predict the true label $y$. However, it will have a catastrophically poor performance on natural test data where $y$-class samples only contain $y$-class features. The overall procedure for generating poisoned data is shown in Algorithm 1.

---

**Algorithm 1** Generating Poisoned Data with Adversarially Inducing Noise

---

**Input:** Source model $f_{\boldsymbol{\theta}_s}$, clean training dataset $\mathcal{D}_c = \{(\boldsymbol{x}_i, y_i)\}$, training steps $M$, poison steps per sample $T$, threshold of fooling success rate $\eta$
**Output:** Poisons $\boldsymbol{\delta}^p$, poisoned training datasets $\mathcal{D}_p$
1: For all $x_i \in \mathcal{D}_c$, randomly initialize a perturbation $\boldsymbol{\delta}^p$ within the $\varepsilon_p$-ball
2: **while** the poison success rate $\text{PSR}(\mathcal{D}_p) \leq \eta$ (Eq. 7) **do**
3:     Update each perturbation $\boldsymbol{\delta}_i^p$ by PGD for $T$ steps using Eq. 5 (with NextCycle by default)
4:     Adversarially train $f_{\boldsymbol{\theta}_s}$ on $\{(\boldsymbol{x} + \boldsymbol{\delta}^p, y')\}$ with inducing labels for $M$ steps (Eq. 6)
5: **end while**
6: **return** Poisoned training dataset $\mathcal{D}_p = \{(\boldsymbol{x}_i + \boldsymbol{\delta}_i^p, y_i)\}$

---

Table 1: The natural and robust test accuracy of the target model, which is trained on poisoned data (generated with different poisoning methods) on CIFAR-10, SVHN, and CIFAR-100.

| Poisoning Methods | CIFAR-10 | | SVHN | | CIFAR-100 | |
|---|---|---|---|---|---|---|
| | Natural | PGD$^{20}$ | Natural | PGD$^{20}$ | Natural | PGD$^{20}$ |
| Clean (baseline) | 85.14% | 51.71% | 91.43% | 56.88% | 57.52% | 27.18% |
| Huang et al. (2021) | 73.42% | 13.34% | **58.54%** | 2.84% | 51.45% | 21.84% |
| Fowl et al. (2021b) | 78.83% | 47.89% | 87.52% | 43.49% | 51.21% | 24.81% |
| ADVIN (ours) | **56.52%** | **0.57%** | 63.88% | **0.46%** | **46.72%** | **11.52%** |

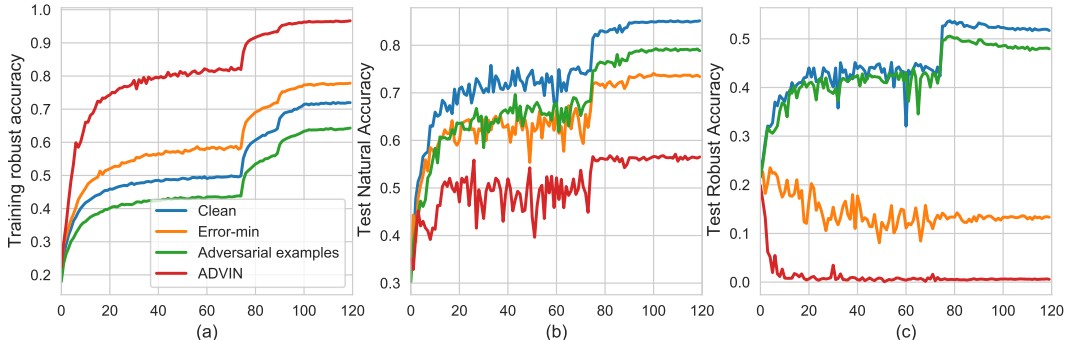

Figure 4: Comparison among baselines (clean data, Huang et al. (2021), Fowl et al. (2021b)) and ADVIN (ours) of robust training accuracy (a), natural test accuracy (b) and robust test accuracy (c) with adversarial training. All experiments are conducted with ResNet-18 on the CIFAR-10 dataset.

## 5 EXPERIMENTS

In this section, we first evaluate our poisoning methods against previous methods on the benchmark datasets and then verify the transferability of our methods across different training algorithms, network architectures, and poisoning ratios. At last, we provide a comprehensive analysis of our ADVIN *w.r.t.* inductive training, target assignments, noise shapes, as well as poisoning threshold.

**Poison Generation.** For the source model used to generate poisons, we adopt the ResNet-18 (He et al., 2016) and train it by AT, where the optimizer is an SGD with a momentum of $0.9$. The initial learning rate is set to $0.1$ and the weight decay is set to $5e^{-4}$. Following Huang et al. (2021), we generate poisons for adversarial training using a relatively large perturbation range $\varepsilon_p = 32/255$[1]. Specifically, we use 60 steps of PGD with step size $2/255$ and generate poisons every 30 training steps until the terminal threshold for PSR is met at $\eta = 0.99$. Besides, we select the NextCycle strategy mentioned in Section 4.2 as our target label mapping function. Typically the poisoning process is very quick and ends within five steps (less than one training epoch). We adopt this setting as default across all our experiments unless specified.

### 5.1 EVALAUTION ON BENCHMARK DATASETS

**Evaluation Protocol.** We evaluate the effectiveness of our poisoning method on fooling AT against previous poisoning methods: error-minimizing (Huang et al., 2021) and adversarial example noise (Fowl et al., 2021b). We conduct experiments on three benchmark datasets, CIFAR-10, SVHN, and CIFAR-100. For each dataset, we adversarially train a ResNet-18 (as the target model) with poisoned data generated with different poisoning methods for 120 epochs. Specifically, we set the initial learning rate as $0.1, 0.1$, and $0.01$ for CIFAR-10, CIFAR-100, and SVHN, respectively. The learning rate decays by $0.1$ at epoch 75, 90, and 100. We adopt an SGD optimizer with a momentum of $0.9$ and a weight decay of $5e^{-4}$. After training, we evaluate the target model on natural (unpoisoned) and adversarial data (PGD$^{20}$ with $\varepsilon_t = 8/255$) and get the natural and robust test accuracy, respectively.

---

[1]As shown in Figure 7 in Appendix D, it will not affect the semantics of the raw images.

Table 2: The natural accuracy and robustness against different AT defense algorithms for ADVIN. Here we conduct experiments on CIFAR-10 with ResNet-18.

| Poisoning Methods | Madry's | | MART | | TRADES | |
|---|---|---|---|---|---|---|
| | Natural | PGD$^{20}$ | Natural | PGD$^{20}$ | Natural | PGD$^{20}$ |
| Clean data | 85.14% | 51.71% | 81.78% | 55.56% | 82.44% | 55.12% |
| ADVIN (ours) | **56.52%** | **0.57%** | **56.40%** | **0.68%** | **56.51%** | **1.66%** |

Table 3: The natural accuracy and robustness of various network architecture for adversarial training. The poisons are generated on CIFAR-10 with ResNet-18 as source model $f_{\boldsymbol{\theta}_s}$

| Poisoning Methods | ResNet-18 | | ResNet-34 | | VGG-11 | | MobileNet-v2 | |
|---|---|---|---|---|---|---|---|---|
| | Natural | PGD$^{20}$ | Natural | PGD$^{20}$ | Natural | PGD$^{20}$ | Natural | PGD$^{20}$ |
| Clean data | 85.14% | 51.71% | 86.21% | 51.64% | 79.05% | 46.60% | 80.42% | 51.01% |
| ADVIN (ours) | **56.52%** | **0.57%** | **56.13%** | **0.54%** | **67.34%** | **4.97%** | **57.53%** | **0.85%** |

**Robustness Drop at Test Time.** We report the performance among clean test datasets of the last epoch in Table 1. For all kinds of datasets, we obtain the lowest robust test accuracy of models trained on ADVIN. Specifically, in terms of robustness, the adversarially trained ResNet-18 have a catastrophic behavior on both CIFAR-10 and SVHN, where the robustness decreases from 51.71% to 0.57% and from 56.88% to 0.46% respectively, showing that ADVIN has severely led the training process disrupted. Besides, on CIFAR-100, Huang et al. (2021) and Fowl et al. (2021b) poisons can only have a slight influence on both natural and robust test accuracy, while we obtain a relative decrease of robust accuracy of 57.6% (Compared to ADVIN, error-minimizing noise and adversarial example noise only obtain a relative reduction of 19.6% and 8.7%, respectively). As for natural test accuracy, we obtain the lowest points on both CIFAR-10 and CIFAR-100. Although error-minimizing noise outperforms our ADVIN by a little margin on SVHN, we still achieve a very low natural accuracy.

**Fooling the Training Process.** Intuitively, since we want to fool the models to learn something from poisoned datasets, we should make sure that models perform well on train datasets and behave badly on test datasets simultaneously. Figure 4 shows the adversarial training process of ResNet-18 on clean examples, error-minimizing noise (Huang et al., 2021), adversarial example noise (Fowl et al., 2021b) and our ADVIN for CIFAR-10 datasets. Figure 4a draws the training robust accuracy curve along training epochs (0 to 120) of four datasets (clean or poisoned). On clean datasets, the ResNet-18 model gets about 70% robust training accuracy while error-minimizing noise and adversarial example noise achieve about 80% and 60% respectively. Surprisingly, the robust training accuracy of ResNet-18 trained on our ADVIN reaches over 95%, which makes the model fully believe that it has been well fitted on the training set, while it will perform very poor on the test datasets as shown in Figure 4b and Figur 4c.

## 5.2 Empirical Understandings

In the above, we have shown the effectiveness of our poisoning method on benchmark datasets when the poisoning stage and the training stage share the same architecture and training objective. However, in practice, if we adopt our poisoning method to protect personal data, the users are agnostic to know how their data will be used for training. Therefore, we further apply our poisons to other training methods and model architectures to study their black-box effectiveness. We also explore whether poisons can still work when they are partially applied. For integrity, we also test the performance of standard training and adversarial training (using different $\varepsilon_t$) on our ADVIN. The results for AT with different training $\varepsilon_t$ and poisoning rate can be found in Appendix B.

**Against Different Defenses.** To valid the generalization among other adversarial training methods, we use TRADES (Zhang et al., 2019) and MART (Wang et al., 2020) as defense algorithms. As shown in Table 2, when training the model with ADVIN, the robustness under PGD$^{20}$ attack will decrease to about 1.7% (trained with TRADES) and 0.7% (trained with MART) respectively, while the natural accuracy remains about 56%. Results show that although the training algorithms of the source model $f_{\boldsymbol{\theta}_s}$ and the target model $f_{\boldsymbol{\theta}_t}$ are different, our poisons can still fool the adversarial training process and make it ineffective.

Table 4: The test accuracy of standard training. Here we conduct experiments on three typical datasets (CIFAR-10, SVHN and CIFAR-100) with ResNet-18. The poisons are generated as described in 5.1.

| Poisoning Methods | CIFAR-10 | SVHN | CIFAR-100 |
|---|---|---|---|
| Clean data | 94.60% | 95.61% | 71.11% |
| ADVIN (ours) | **13.12%** | **9.15%** | **2.39%** |

**Transferability Across Architectures.** Intuitively, the poisoned datasets should also destruct the training process whatever the network architectures are used during training. To valid the transferability, we adversarially train our ADVIN on different network architectures. Table 3 shows that poisoned data crafted by ResNet-18 could also transfer to other models. The performance on ResNet-34 and MobileNet-v2 is almost equal to the performance on ResNet-18, with natural accuracy decreasing to about 57% and robust accuracy decreasing to lower than 1% respectively.

**Effectiveness on Standard Training.** Since ADVIN has shown the effectiveness of fooling various adversarial training methods, it is reasonable that ADVIN can even disrupt the standard training process more thoroughly. Therefore, we replace the adversarial training for the target model with standard training. For all the three datasets, we train them for 60 epochs with an initial learning rate of 0.1. An SGD and a MultiStepLR are used for optimization. Table 4 reports the accuracy of ResNet-18 trained on benchmark datasets, which demonstrates that our ADVIN can destroy the performance of models and lead the accuracy close to random guess (*e.g.,* 9.15% accuracy for SVHN).

## 5.3 PARAMETER ANALYSIS

Here, we provide a thorough analysis of the generation process of our proposed inducing noise in terms of the following aspects. First, we show that adversarial training is necessary in our inductive training process by comparing it with standard training. Then, we show our poisoning is still effective when being applied to a small region in the image. Besides, in Appendix C, we study the effect of different adversarial training algorithms for source models and showing that they are all useful for various defense algorithms. We also analyze the effect of alternative label assignments and PSR threshold. Results show our method is relatively robust to these choices.

**Adversarial Inducing Noise v.s. Standard Inducing Noise.** As discussed in Algorithm 1, generating poisons by ADVIN is an iterative process that could be divided into two stages, adversarial training and noise generation with the source model. Here, we replace the adversarial training with standard training instead and name the poisons as STanDard Inducing Noise or STDIN. Note that in contrast to Fowl et al. (2021b), we train the source model from scratch instead of using a pre-trained model. As shown in Figure 5, although STDIN achieves higher robust training accuracy than clean data, there is still a big gap between STDIN and our ADVIN, which indicates that ADVIN can fool the source model more thoroughly. As for the poisoning effect on the test data, STDIN is also effective to some extent, with a natural test accuracy dropping by about 16% and robust test accuracy dropping by about 46%. Nevertheless, our ADVIN still outperforms STDIN by a large margin (13% natural test accuracy and 5% robust test accuracy). This comparison illustrates the effectiveness of adopting adversarial training for generating poisons in our ADVIN.

**Effectiveness of Small-size Poisons.** In previous experiments, we apply the noise to full images, while here, we explore the effectiveness of small-size noise. We expect the smaller-patched noise to fool the training of target models like triggers and at the same time retain semantic information as much as possible. We choose 8*8, 16*16, and 24*24 as patched sizes. As expected, they can all destroy AT as shown in Table 5. Surprisingly, when the model is trained on 16*16 patched noise, it gains the lowest test natural accuracy of 44.16%, which is about 12% lower than 32*32 patched noise, and second-lowest test robust accuracy of 0.76%, which is also less than 1%.

## 6 REAL-WORLD APPLICATION FOR DATA PRIVACY PROTECTION

As introduced above, we can utilize our poisoning to protect personal data from being exploited by commercial companies. Here we consider a real-world scenario where personal profile photos online could be crawled down for training face recognition systems without permission. Below, we show that adding our poisoning to the data can successfully protect the data from being exploited by either standard or adversarial training and make them truly unlearnable examples.

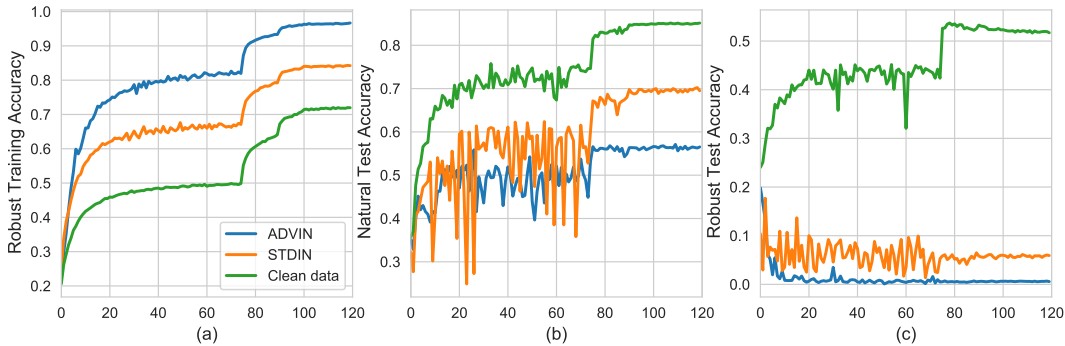

Figure 5: Comparison between ADVersarial Inducing Noise (ADVIN) and STanDard Inducing Noise (STDIN) of robust training accuracy (a), natural test accuracy (b) and robust test accuracy (c) with adversarial training. All experiments are conducted with ResNet-18 on the CIFAR-10 dataset.

Table 5: The natural accuracy and robustness of ResNet-18 for AT. The poisons are generated on CIFAR-10 with ResNet-18 as source model $f_{\theta_s}$. We set the shape of noise to 8*8, 16*16, 24*24 and 32*32 (the size of full images) respectively.

| 8*8 | | 16*16 | | 24*24 | | 32*32 | |
|---|---|---|---|---|---|---|---|
| Natural | PGD$^{20}$ | Natural | PGD$^{20}$ | Natural | PGD$^{20}$ | Natural | PGD$^{20}$ |
| 72.19% | 19.69% | **44.16%** | 0.76% | 66.93% | 6.49% | 56.52% | **0.57%** |

**Setup.** We choose Webface as our raw dataset, which includes about 490k images of over 10k identities. For simplicity, we select the ten most frequent classes of images as our sub-dataset and name it Webface-10. The Webface-10 dataset consists of 5338 images for training and 1340 images for testing. Specifically, we want to protect the selected sub-datasets and generate noise for them, which leads the models to get fooled by the noise. Therefore, on both clean and poisoned Webface-10, we train a ResNet-18 where we set the learning rate to $0.01$ and weight decay to $5e^{-4}$ for ST (60 epochs) and AT (120 epochs). Also, we choose CosineAnnealingLR as the scheduler of ST, and the learning rate drops by $0.1$ for ST. An SGD with a momentum of $0.9$ is used for optimization.

Table 6: Both the accuracy under standard training and the natural/robust test accuracy under adversarial training for Webface-10. We use ResNet-18 for both the source model and the target model.

| Poisoning Methods | Natural acc (ST) | Natural acc (AT) | Robust acc (AT) |
|---|---|---|---|
| Clean data | 89.18% | 81.34% | 43.81% |
| ADVIN (ours) | **27.46%** | **40.82%** | **22.69%** |

**Results.** As shown in Table 6, the natural accuracy could reach $89\%$ and $81\%$ with standard training and adversarial training, respectively. In comparison, with our poisoned data, standard training could only obtain $27.46\%$ natural accuracy, and adversarial training can only achieve $40.82\%$ natural accuracy and $22.69\%$ robust accuracy. This shows that our poisons can actually protect the users' data from being mined and utilized for training.

## 7  CONCLUSION

In this paper, we have designed a new kind of poisoning method, Adversarial Inducing Noise (ADVIN), for fooling adversarial training. Extensive experiments on a range of benchmark datasets show that the generated poisons can make adversarial training ineffective no matter what different training strategies and model architectures are adopted. Besides, we can conduct a thorough analysis of the poisoning generating process, showing that our poisoning is effective under different stopping criteria and (fixed) label assignment strategies. At last, we successfully apply our method to protecting personal data privacy against adversarial training on face recognition.

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

## A    EXPERIMENTAL DETAILS

### A.1    DETAILS OF WHY ERROR-MINIMIZING NOISE FAILS

We adversarially and standardly train ResNet-18 with clean datasets and error-minimizing poisoning datasets for CIFAR-10, respectively. Specifically, we set $\varepsilon_p$ to $8/255$ for error-minimizing noise and run the command released at https://github.com/HanxunH/Unlearnable-Examples. For standard training, we use an SGD with a momentum of $0.9$ and set the learning rate to $0.1$ with weight decay $5e^{-4}$. The learning rate drops through a CosineAnnealingLR scheduler. For training and testing, we all set the batch size to $128$. The setting of adversarial training is very close, except we train the model for 120 epochs, and the learning rate drops by $0.1$ at epoch 75, 90 and 100. In addition, we set the perturbation budget of adversarial training $\varepsilon_t$ to $8/255$.

### A.2    DISCUSSION ABOUT ROBUST AND NON-ROBUST FEATURES

We first standardly and adversarially train a ResNet-18 model as the source model, and then use PGD attack to generate adversarial examples as poisons. The standardly pre-trained ResNet-18 is trained for 60 epochs with an initial learning rate of $0.1$, weight decay $5e^{-4}$, and a CosineAnnealingLR as the scheduler, while we train the robust ResNet-18 model by AT (Madry et al., 2017) for 120 epochs with a MultiStepLR. The learning rate drops by $0.1$ at epoch 75, 90, and 100. For both adversarial training and standard training, we set the training batch size to $128$. An SGD with a momentum of $0.9$ is used for optimization. For noise generation, we use targeted PGD$^{200}$ with step size $2/255$ to generate adversarial examples as poisons. In terms of targeted attack, we choose NextCycle as mentioned in Section 4.2 for label mapping.

### A.3    EXPERIMENTS OF LABEL CONSISTENT BIAS

As Section 4.1 has discussed, we use an adversarially pre-trained ResNet-18 model to generate adversarial examples as poisons. The process of training a ResNet-18 source model and poison generation is just the same as A.2, except that we use five different target label mapping functions (Random,LL, MC, NextCycle, and NearSwap).

## B    EFFECTIVENESS ACROSS DIFFERENT TRAINING SETTINGS

**Results for AT of different training** $\varepsilon_t$**.** Intuitively, the poisons generated should also fool the models adversarially trained with different $\varepsilon_t$ (*e.g.,* $2/255, 4/255, 8/255, 16/255$ and $32/255$). Therefore, we train a ResNet-18 on poisoned CIFAR-10 with AT under different $\varepsilon_t$. From Table 7, our ADVIN can decrease the performance among all the models trained with different $\varepsilon_t$, especially when the $\varepsilon_t$ is less and equal to $16/255$. For example, whether the models are adversarially trained with $\varepsilon_t = 2/255$ or $4/255$, they entirely lose the robustness and could be easily attacked by PGD$^{20}$. Besides, the natural test accuracy (about $20\%$) is also much lower than the results on clean data (about $90\%$). While the models are trained with $\varepsilon_t = 32/255$, our ADVIN could also slightly affect the robust and natural test accuracy (about a drop of $2\%$). The effectiveness of our ADVIN seems to

Table 7: The natural accuracy and robustness of AT under different $\varepsilon_t$. The models are trained on poisoned CIFAR-10, which are generated with ResNet-18 as source model $f_{\boldsymbol{\theta}_s}$

| Poisoning Methods | 2/255 | | 4/255 | | 8/255 | | 16/255 | | 32/255 | |
|---|---|---|---|---|---|---|---|---|---|---|
| | Natural | PGD$^{20}$ | Natural | PGD$^{20}$ | Natural | PGD$^{20}$ | Natural | PGD$^{20}$ | Natural | PGD$^{20}$ |
| Clean data | 92.62% | 29.26% | 90.17% | 41.05% | 85.14% | 51.71% | 67.94% | 52.54% | 34.50% | 29.38% |
| ADVIN (ours) | **18.05%** | **0%** | **22.16%** | **0%** | **56.52%** | **0.57%** | **67.36%** | **44.25%** | **32.53%** | **27.83%** |

Table 8: The natural accuracy and robustness under partially poisoned datasets $\mathcal{D}_p + \mathcal{D}_c$ and part of clean datasets $\mathcal{D}_c$ for adversarial training. The poisons are generated on CIFAR-10 with ResNet-18.

| Dataset | 0% | | 20% | | 40% | | 60% | | 80% | | 100% | |
|---|---|---|---|---|---|---|---|---|---|---|---|---|
| | Natural | PGD$^{20}$ | Natural | PGD$^{20}$ | Natural | PGD$^{20}$ | Natural | PGD$^{20}$ | Natural | PGD$^{20}$ | Natural | PGD$^{20}$ |
| $\mathcal{D}_c$ | 85.14% | 51.71% | 84.05% | 49.51% | 82.61% | 46.67% | 79.03% | 42.64% | 73.98% | 35.15% | – | – |
| $\mathcal{D}_c + \mathcal{D}_p$ | – | – | 84.69% | 51.64% | 84.25% | 48.68% | 82.57% | 44.47% | 80.57% | 37.92% | 56.52% | 0.57% |

be reduced with $\varepsilon_t = 32/255$. This may be due to the fact that the noise could be distorted by the training perturbation with the same order of magnitude as ADVIN.

**Poisoning Rate.** In the practice scenario, it is very likely that not all the training examples are poisoned as mentioned in Huang et al. (2021). Therefore we randomly select a certain portion of data to add noise while keeping the remaining training examples clean. The partially poisoned datasets are denoted as as $\mathcal{D}_p + \mathcal{D}_c$. We adversarially train the poisoned datasets with different poison rates. In comparison, we also use the subset of clean training data for adversarial training, where the portion of clean examples is the same as partially poisoned datasets $\mathcal{D}_p + \mathcal{D}_c$, and we denote it as $\mathcal{D}_c$. Results show that a small portion of clean examples could help DNNs get trained well and lead to a huge increase both on test robust and natural accuracy as shown in Table 8. For example, when we select $80\%$ of poisoned examples and $20\%$ clean data jointly as $\mathcal{D}_p + \mathcal{D}_c$. After we adversarially train the mixed datasets, we gain a natural and robust test accuracy of $80.57\%$ and $37.92\%$, which are even a little higher than the models trained on only $\mathcal{D}_c$. A similar result can be observed in previous work (Huang et al., 2021) and (Shan et al., 2020).

# C  ANALYSIS ON POISON GENERATION

## C.1  INDUCING ADVERSARIAL TRAINING WITH OTHER AT VARIANTS

Here we utilize MART (Wang et al., 2020) and TRADES (Zhang et al., 2019) for training the source model. For a fair comparison, we also evaluate the target model trained by MART and TRADES instead. As expected, from Table 9, we can see that the poisons crafted by other AT variants could also generate useful noise. Especially the noise generated by TRADES could lead the natural test accuracy even lower than $50\%$ whatever the training algorithms for the target model are. In comparison, ADVIN could only lead the models to reach a natural test accuracy of about $56\%$.

## C.2  INDUCING TARGET LABEL

From the discussion of Section 4.2, we know that a consistently target label bias is critical for noise generation. We replace the label mapping function mentioned in Algorithm 1 (We use NextCycle strategy as default for previous experiments.) and test the ability to disrupt the training process of the noise with other target label induction. Table 10 lists the test robust and natural accuracy of ResNet-18 on CIFAR-10. The models are all trained on ADVIN but with different targeted inducing label mapping. NearSwap denotes the swap between the nearest classes as mentioned in Section 4.2, while SimilarSwap and DissimilarSwap indicate the swap between semantically similar classes and semantically dissimilar classes. Motivated by Tian et al. (2021), we use the confusion matrix to estimate the semantic similarity. Here we also want to explore the class-wise property of the datasets. Since we assign every class an induced label during noise generation, the poisoned training examples should include the features of induced classes to some extent, which may lead to the results that when we test the clean datasets, the predicted results should show some bias to the reversely induced label (*e.g.,* when training we use class 2 as the inducing label for class 1 under NextCycle setting, then during the test time, the predicted results of class 1 should have a tendency towards class 2).

Table 9: Poisons are generated from source models trained with different methods. Here lists the performance of target models trained with Mardy's (Madry et al., 2017), MART (Wang et al., 2020) and TRADES (Zhang et al., 2019). Rows correspond to the training methods for source models, and cols correspond to the training methods for target models.

| Defense / Poisoning | Madry's | | MART | | TRADES | |
|---|---|---|---|---|---|---|
| | Natural | PGD$^{20}$ | Natural | PGD$^{20}$ | Natural | PGD$^{20}$ |
| Madry's | 56.52% | **0.57%** | 56.40% | **0.68%** | 56.51% | **1.66%** |
| MART | 52.83% | 7.05% | 50.20% | 7.54% | 53.44% | 11.73% |
| TRADES | **44.87%** | 1.49% | **41.99%** | 1.61% | **46.18%** | 3.90% |

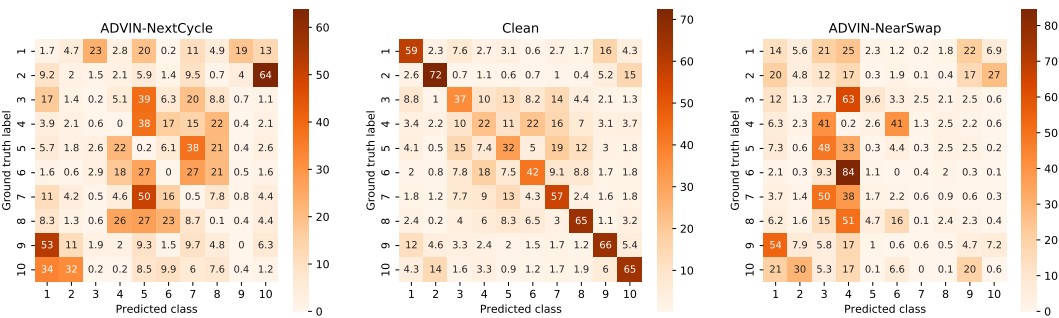

Figure 6: Confusion matrix for clean CIFAR-10 test datasets of models trained with different poisons(ADAIN with NextCycle label assignment, Clean training data, and ADAIN with NearSwap label assignment, respectively.) where rows correspond to the ground truth labels and cols correspond to the predicted labels.

Figure 6 shows the confusion matrix under clean and poisoned data for CIFAR-10. From that, we can see whether under NextCycle or NearSwap setting, the misclassified labels represent a certain degree of consistency compared with the confusion matrix under clean datasets. For example, under clean data setting, adversarial examples of class 9 have a probability of 12% to be predicted to class 1, while under ADVIN of NextCycle and NearSwap, the probability increase up to 53% and 54% respectively. Besides, we indeed observe that the induced labels have some effects on the predicted classes. The adversarial examples of class 10 tend to be predicted as class 1 under the NextCycle setting, where they have a tendency towards class 9 under the NearSwap setting.

## C.3 STOPPING CRITERION

Recall that the signal for terminating the loops of noise generation is $\text{PSR}(\mathcal{D}_p)$ surpasses the threshold of fooling success rate $\eta$. Here we discuss the influences of different threshold $\eta$ on the effectiveness of poisons. Specifically, we choose 90%, 99%, and 99.9% as rate threshold to control the termination for noise generation. As shown in Table 11, when we set the threshold to 90%, the natural and robust test accuracy achieve about 64% and 8%, respectively. Compared to the results when the threshold is set to 99% and 99.9%, it only gets about 8% improvement. In general, although the performance of models trained on noise with different thresholds has some slight differences, the effectiveness of ADVIN is insensitive to the $\text{PSR}(\mathcal{D}_p)$ threshold.

## D VISUALIZATION OF NOISE GENERATED FROM PRE-TRAINED MODEL

We randomly select part of images of CIFAR-10 and generate robust features as noise from a adversarially pre-trained model as mentioned in 4.1. Here we compare the poisoned images with noise under different $\varepsilon_p$. From Table 7 we can see that although the $\varepsilon_p$ is increased to $32/255$, the poisoned images could still remain semantic information well.

Table 10: The natural accuracy and robustness of ResNet-18 for adversarial training. The poisons are generated on CIFAR-10 with ResNet-18 as the source model $f_{\theta_s}$, where four kinds of inductive label mapping function are utilized as mentioned in Algorithm 1.

| NextCycle | | NearSwap | | SimilarSwap | | DissimilarSwap | |
|---|---|---|---|---|---|---|---|
| Natural | PGD$^{20}$ | Natural | PGD$^{20}$ | Natural | PGD$^{20}$ | Natural | PGD$^{20}$ |
| 56.52% | 0.57% | 54.92% | 2.96% | 63.69% | 7.92% | 55.91% | 5.95% |

Table 11: The natural accuracy and robustness under AT, with being trained on ADVIN generated from different fooling success rate $\eta$. The experiments are conducted on CIFAR-10 with ResNet-18.

| PSR($\mathcal{D}_p$) threshold | 90% | | 99% | | 99.9% | |
|---|---|---|---|---|---|---|
| | Natural | PGD$^{20}$ | Natural | PGD$^{20}$ | Natural | PGD$^{20}$ |
| ADVIN | 64.21% | 8.44% | 56.52% | 0.57% | 56.52% | 0.57% |

## E  SUPPLEMENTARY DISCUSSION ABOUT RELATED WORK

Feng et al. (2019) first propose the data poisoning problem where training data are allowed to be modified with a bounded perturbation such that the DNNs trained on the poisoned data could have a poor performance on clean test data. However, Feng et al. (2019) adopt auto-encoder to generate poisons $\delta_p$ under a white-box setting with full access to the targeted model, making them not practical in real-world scenario. Fowl et al. (2021a) instead achieve this by aligning the training gradient of the perturbed data with the gradient of the reverse cross-entropy loss on the clean data, and Yuan & Wu (2021) propose Neural Tangent Generalization Attacks based on Neural Tangent Kernels (Jacot et al., 2018). Besides, Evtimov et al. (2021) explore three class-wise patterns of shortcuts which control the trade-off between disrupting training and preserving visual features in the data. While these work focus on poisoning standard training, Tao et al. (2021) theoretically show that adversarial training is an effective method to defend these attacks. Different from theirs, the goal of this work is to develop a poison algorithm that works on both adversarial training as well as standard training.

## F  MORE POISONED IMAGES

To show that a poison budget $\varepsilon_p = 32/255$ will not modify the clean labels, we randomly select ten poisoned images from each class in CIFAR-10 and compare them with clean ones. The poisons are generated through an adversarially pre-trained model as described in Section 4.1. Note that here we set the poison budget $\varepsilon_p = 32/255$. As shown in Figure 8, the poisons will not affect the semantics of images nor modify the true labels.

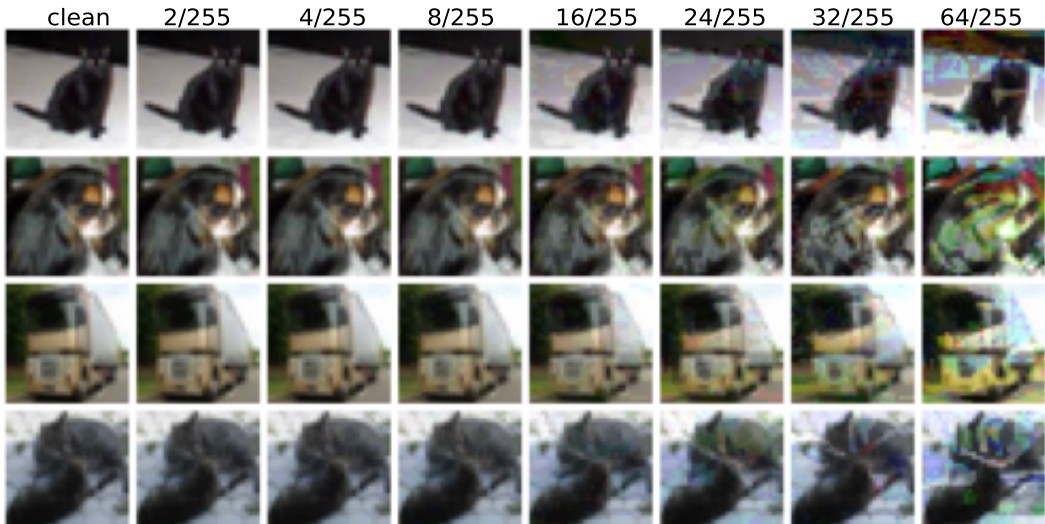

Figure 7: The noise generated from PGD$^{200}$ attack with a pre-trained robust model under different $\varepsilon_p$-ball.

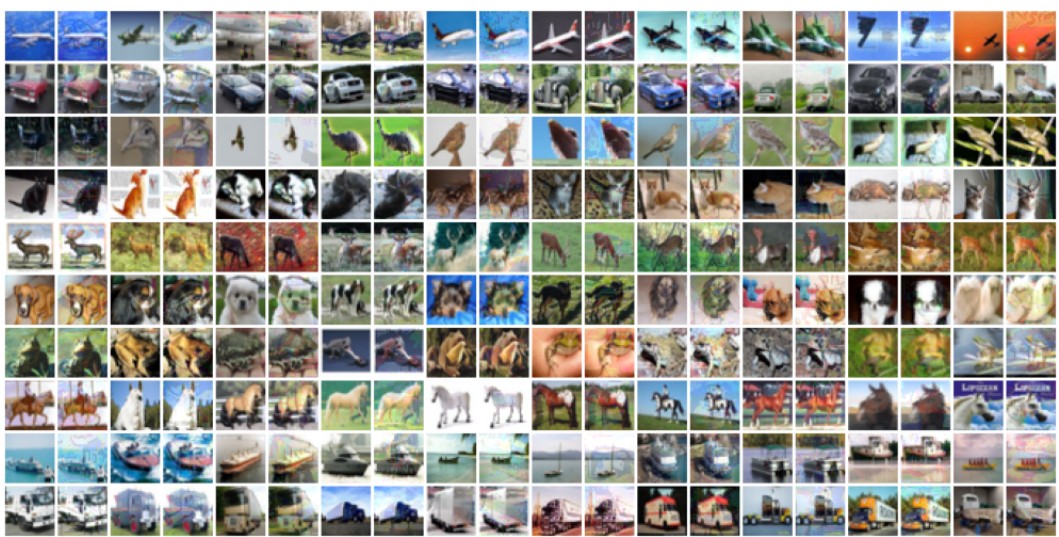

Figure 8: The poisoned images generated from PGD$^{200}$ attack with a pre-trained robust model. Ten images are randomly selected from each of the class in CIFAR-10. Each row shows ten pairs of clean and poisoned images. Here the poison budget $\varepsilon_p$ is set to $32/255$.

