# OpenReview forum: "Fooling Adversarial Training with Induction Noise"
_ICLR.cc/2022/Conference — ICLR 2022 Submitted_

### Official Review · Reviewer_r3Un · 2021-10-30

**Correctness:** 4
**Technical Novelty And Significance:** 3
**Empirical Novelty And Significance:** 3
**Recommendation:** 6
**Confidence:** 4

**Main Review:**

My concern for this paper is along the following lines: The threat model is really not clear -- and to the degree of incomparable.

For example, when the paper first discusses data poisoning for standard training by Huang et al. (3), that threat model is about to say
that we do not modify the labels, but very small perturbations on the features **suffices** to fool the model. The fact that labels are not allowed to modify is the whole point, and the actual interesting part, of Huang's approach.

But this paper then silently switches to allow labels to be changed -- see (4). This is already a significant change of threat model, and if labels are allowed to change, why would it be surprising that AT can be fooled? In fact, if labels are allowed to change, then intuitively AT can be in an even worse situation than standard training: This is because that AT will insist to learn w.r.t. a wrong label, because we want "robustness". This seems to really diminish the "interestingness" of this paper.

I understand that this paper investigated a somewhat deep issue of consistent label bias. This is interesting -- and as I said this paper has some interesting content. To this end, I feel this paper at least needs a significant revision to make clear how to compare with previous work, and why the studied threat model is interesting after all.

**Summary Of The Paper:**

This paper studies a data poisoning method for adversarial training. I like this paper in general: The derivations are clear and empirical results are clearly presented. However, this paper may suffer from unclear formulations of threat model, as I will discuss below in more detail.

**Summary Of The Review:**

As I mentioned in the detailed review, I think this paper silently changed the threat model and allow modification of labels -- this is a significant change, and actually puts AT to a worse situation compared to ST (at least there are intuitions that would support such an argument). More revisions are needed -- in fact I think a significant revision is needed.

---

> ### Author Response · Authors · 2021-11-19
> **Response to Reviewer r3Un**
>
> Thanks sincerely for appreciating the novelty of our work. However, they might be some misunderstanding of our poisoning setup, as we explain below.
>
> ---
>
> **Q1.** I think this paper silently changed the threat model and allow modification of labels -- this is a significant change.
>
> **A1.** We are afraid there are some understandings here. Our problem setup is the **same as Huang et al. [1]**, where we are allowed to add small perturbations to the input (as poisons) while keeping the labels unchanged, leading to the poisoned data ${D}_p=\{(x_i+\delta_i^p, y_i)\}$.
>
> Our difference merely lies in **the poisoning process**, i.e., how to generate the perturbations $\{\delta_i^p\}$ with a source model $f_{\theta_s}$ for every input. To train this source model, we adopt a label assignment strategy that deliberately induces the perturbed input $x_i+\delta_i^p$ to the target label $y'$. Notably, here we only use the source model  $f_{\theta_s}$ trained in this way to generate the input perturbations $\{\delta_i^p\}$, but **not modify the labels of the poisoned data** $D_p$ that are used for training the target model $f_{\theta_t}$. That being said, because the source model is owned by the attacker for generating input poisons, we can surely modify the training objective freely, including modifying labels.
>
> [1] Hanxun Huang, Xingjun Ma, Sarah Monazam Erfani, James Bailey, Yisen Wang. Unlearnable Examples: Making Personal Data Unexploitable. In ICLR. 2021.
>
> ---
>
> Thanks again for your review and hope our explanations could address your concerns. Please let us know if you have additional questions.

---

> > ### Comment · Reviewer_r3Un · 2021-11-19
> > **Ack your reply**
> >
> > Thank you for correcting my mistake. I've updated my score to 6, and I think this is an interesting paper.
> >
> > I will talk to other reviewers and potentially update my score further.

---

### Official Review · Reviewer_Hbey · 2021-10-30

**Correctness:** 3
**Technical Novelty And Significance:** 3
**Empirical Novelty And Significance:** 4
**Recommendation:** 6
**Confidence:** 4

**Main Review:**

Pros:

- This paper is the first to study the problem of poisoning the performance of adversarial training.
- The proposed poisoning attack can successfully make adversarial training with small $\epsilon$ ineffective.

Cons:

- This paper lacks an adequate discussion/reference of a few closely related works, including [1], [2], [3], [4], and [5]. In short, they all studied the threat that aims to degrade test accuracy by perturbing training data.
- The main drawback of this paper is that using such a large perturbation range $\epsilon_p=32/255$ for generating poisons may be not so reasonable. Actually, The commonly used $\ell_{\infty}$ perturbation budget in the literature is 0.3 for MNIST and 8/255 for CIFAR-10, SVHN, and CIFAR-100. However, it has been shown in [6] that perturbations with a large budget for MNIST may modify an input’s true label. Though [6] only conduct experiments on MNIST, the same conclusion may be true for other commonly used datasets. The few examples in Figure 7 are not enough to support the claim "it will not affect the semantics of the raw images".
- In section 4.1, could you tell me the budget of training perturbations for adversarial training?
- It may not be suitable to claim the success of the poisons when the the budget of training perturbations $\epsilon_t=8/255$ is much small than the poison budget $\epsilon_p=32/255$, since adversarial training needs a large $\epsilon$ to guarantee its performance (cf. Theorem 1 in [2]).
- As a reminder, [6] was the first to point out that adversarial example noise can be used as poisons. Later, this idea was experimented in [2] and [7].
- As a reminder, on Page 10, the reference to MART appears twice.

[1] Feng et al., Learning to Confuse: Generating Training Time Adversarial Data with Auto-Encoder, NeurIPS 2019
[2] Tao et al., Better Safe Than Sorry: Preventing Delusive Adversaries with Adversarial Training, NeurIPS 2021
[3] Fowl et al., Preventing Unauthorized Use of Proprietary Data: Poisoning for Secure Dataset Release, arxiv preprint, 2021
[4] Yuan et al., Neural Tangent Generalization Attacks, ICML 2021
[5] Evtimov et al., Disrupting Model Training with Adversarial Shortcuts, ICML 2021 Workshop
[6] Nakkiran, Adversarial Examples are Just Bugs, Too, Distill, 2019. https://distill.pub/2019/advex-bugs-discussion/response-5/
[7] Fowl et al., Adversarial examples make strong poisons, NeurISP 2021

**Summary Of The Paper:**

This paper proposes to reduce the adversarial robustness of adversarial training by perturbing the training data with large perturbation budget. Various strategies for assigning fooling labels are explored, which eventually leads to a strong poisoning attack named IAT. Extensive experiments show the effectiveness of the proposed poisoning algorithm.

**Summary Of The Review:**

Overall, I thought the threat model under consideration is interesting and worth exploring, but the authors may need to explain more motivations for their choice of threat budget. I would increase the rating if an updated draft addresses the mentioned issues in the main review.

---

> ### Author Response · Authors · 2021-11-19
> **Response to Reviewer Hbey (2/2)**
>
> **Q5.** Reference issues: a reminder of related work and a reference mistake.
>
> **A5.** Thanks for pointing them out. We have added relevant discussions in our paper as you suggested, and have deleted the first duplicated entry in the updated version.
>
> ---
>
> Thanks again for your constructive review, and please let us know if you have further questions.

---

> ### Author Response · Authors · 2021-11-19
> **Response to Reviewer Hbey (1/2)**
>
> Thanks sincerely for your constructive comments and highlighting the potential impact of our work. We address your concerns as follows.
>
> ---
>
> **Q1.** Lack of adequate discussion of related work.
>
> **A1.** Thanks for pointing out those related work. We have added the following discussions in **Appendix E** following your suggestions:
> > Feng et al. [1] first propose the data poisoning problem where training data are allowed to be modified with a bounded perturbation such that the DNNs trained on the poisoned data could have a poor performance on clean test data. However, Feng et al. [1] adopt auto-encoder to generate poisons $\delta_{p}$ under a white-box setting with full access to the targeted model, making them not practical in real-world scenario. Fowl et al. [2] instead achieve this by aligning the training gradient of the perturbed data with the gradient of the reverse cross-entropy loss on the clean data, and Yuan et al. [3] propose Neural Tangent Generalization Attacks based on Neural Tangent Kernels [4]. Besides, Evtimov et al. [5] explore three class-wise patterns of shortcuts which control the trade-off between disrupting training and preserving visual features in the data. While these work focus on poisoning standard training, Tao [6] theoretically show that adversarial training is an effective method to defend these attacks. Different from theirs, the goal of this work is to develop a poison algorithm that works on both adversarial training as well as standard training.
>
> Reference:
>
> [1] Ji Feng, Qi-Zhi Cai, Zhi-Hua Zhou. Learning to Confuse: Generating Training Time Adversarial Data with Auto-Encoder. In NeurIPS.  2019
>
> [2] Liam Fowl, Ping-yeh Chiang, Micah Goldblum, Jonas Geping, Arpit Bansal, Wojtek Czaja, Tom Goldstein. Preventing Unauthorized Use of Proprietary Data: Poisoning for Secure Dataset Release. Arxiv preprint. 2021
>
> [3] Chia-Hung Yuan, Shan-Hung Wu. Neural Tangent Generalization Attacks. In ICML. 2021
>
> [4] Arthur Jacot, Franck Gabriel, Clément Hongler. Neural Tangent Kernel: Convergence and Generalization in Neural Networks. In NeurIPS. 2018.
>
> [5] Ivan Evtimov, Ian Covert, Aditya Kusupati, Tadayoshi Kohno. Disrupting Model Training with Adversarial Shortcuts. In ICML Workshop. 2021
>
> [6] Lue Tao, Lei Feng, Jinfeng Yi, Sheng-Jun Huang, Songcan Chen. Better Safe Than Sorry: Preventing Delusive Adversaries with Adversarial Training. In NeurIPS. 2021.
>
> ---
>
> **Q2.** A few examples might not be enough to support the claim that a perturbation range of $32/255$ will not affect the semantics of the raw images.
>
> **A2.** Thanks for pointing out this issue. To support the claim, we randomly select ten poisoned images from all the ten classes of CIFAR-10. Although they have been poisoned with a perturbation of range $\varepsilon_p=32/255$, the semantics of those poisoned images is remained. The poisoned images could be found here in **Appendix F**.
>
> ---
>
> **Q3.** The budget of training perturbations for adversarial training.
>
> **A3.** Sorry, we miss it for the experiments in **Section 4.1**. We set the range of perturbation to $\varepsilon_t=8/255$. We have added this information to **Appendix A.1**.
>
> ---
>
> **Q4.** The issue that the budget of poisons ($\varepsilon_p=32/255$) is larger than the budget of adversarial training ($\varepsilon_t=8/255$).
>
> **A4.** **First**, as discussed in **Appendix B**, we conduct experiments of adversarial training under a large budget $\varepsilon_t=16/255$, and $\varepsilon_t=32/255$, respectively. As shown in **Table 7**  (quoted below), our ADVIN can decrease the robustness of about $8\%$ and $2\%$ when the training budget $\varepsilon_t$ is set $16/255$ and $32/255$. In other words, no matter how large the adversarial training budget $\varepsilon_t$ we choose, ADVIN can always effectively reduce the robustness of DNNs.
>
> | Training budget $\varepsilon_t$ | 2/255   |        |  4/255  |        | 8/255   |        | 16/255  |        | 32/255  |        |
> | ------------------------------- | ------- | ------ | :-----: | ------ | ------- | ------ | ------- | ------ | ------- | ------ |
> |                                 | Natural | PGD    | Natural | PGD    | Natural | PGD    | Natural | PGD    | Natural | PGD    |
> | Clean data                      | 92.62%  | 29.26% | 90.17%  | 41.05% | 85.14   | 51.71% | 67.94%  | 52.54% | 34.50%  | 29.38% |
> | ADVIN                           | 18.05%  | 0%     | 22.16%  | 0%     | 56.52%  | 0.57%  | 67.36%  | 44.25% | 32.53%  | 27.83% |
>
> **Second**, we also conduct experiments of training of clean data. **Table 7** shows that a larger training budget of $16/255$ or $32/255$ will affect the robustness and accuracy whether the noise is injected into the clean data or not. And ADVIN could severely decrease the performance of adversarial training under a budget $\varepsilon_t$ less and equal to $8/255$, which means our ADVIN can always make poisoned data unlearnable no matter how defender selects the training budget $\varepsilon_t$.

---

> ### Author Response · Authors · 2021-11-29
> **Need further clarification?**
>
> Thanks for your constructive comments. We have tried our best to address the concerns. Is there any unclear point that we should/could further clarify?

---

> > ### Comment · Reviewer_Hbey · 2021-11-29
> > **Thanks for the response**
> >
> > Thanks for your detailed response and the revision. Most of my concerns have been addressed. I hope the authors could also carefully address the concerns raised by the other reviewers, e.g., the novelty issue and the detection issue caused by large perturbation budge (Reviewer eute).

---

> > > ### Author Response · Authors · 2021-11-30
> > > **Thanks**
> > >
> > > Thanks for your kind response and appreciating our feedbacks.
> > >
> > > As for your additional concerns, we have 1) clarified the novelty issue through a careful comparison with related work; and 2) further added the detection results in the recent response to Reviewer eute. Please take a look and hope it could address your concerns.

---

### Official Review · Reviewer_H39o · 2021-11-02

**Correctness:** 3
**Technical Novelty And Significance:** 3
**Empirical Novelty And Significance:** 3
**Recommendation:** 5
**Confidence:** 4

**Main Review:**

The work is well written. It presents an effective approach to performing such poisoning; however, the work does not sufficiently explore the reasons for the success of the technique (ADVIN) or compare it to appropriate baselines. The work could also improve clarity in some areas.

Fleshing out technique/baselines: ADVIN, the main presented technique of the work, is not explained or motivated sufficiently. It is difficult to make sense of why this technique works, or even why it should work to begin with---I would have thought that changing the model objective so much would make the network relatively useless when poisoning new adversarially trained models. It would be good to start out with simple extensions of the AT-Pretrained method first: what happens when you ensemble and do the AT-Pretrained method, or adversarially train for the standard (i.e. proper labels with their corresponding perturbed $x$ images) objective in the loop?

Clarity:
- It would be good to expand on Huang et al 2021 around equation (3) for greater context on methods used in this area (especially considering that the techniques in the work heavily build upon Huang et al 2021).
- Minor point: the title of the forum page for this paper does not match the title of the pdf of this paper.

**Summary Of The Paper:**

This work tackles the problem of data poisoning adversarial training in the (image classification) setting where the adversary can add $\ell_p$ constrained perturbations to each training example. Others have previously studied this problem in the standard ERM setting by adding $\ell_p$ perturbations to examples that minimize loss on pretrained models: when the downstream learner tries to minimize risk on these perturbed examples, they can do well on the train set but not on a held-out set from the original distribution (this work is all performed in CIFAR). It turns out that adversarially trained models can largely circumvent this class of attacks, as they can modify the adversarial $\ell_p$ perturbations in the adversarial training process to make learning require appropriately generalizable features.

The authors try a new attack that focusses on adding "robust" features to examples: in the most basic attack, for each (example, label) combination the authors perturb the example to maximize the probability of a different (label-consistent --- i.e. each image of a given label is perturbed towards the same class, one such mapping could be target_label = label + 1 % num_classes ) label. The authors specialize the attack further by performing an alternating optimization routine in which one repeatedly (starting from a pretrained adv-trained model $f$): (a) completes the perturbation above on the examples using $f$ and then (b) adversarially trains on $f$ using the new label scheme. Using this technique the authors can force the model to perform well on the train set, but not on a held-out set. Using only step (a) (i.e. no alternating optimization routine) the authors can fool the model in the same sense but to a lesser extent.

**Summary Of The Review:**

The work presents a new technique that effectively poisons adversarially trained models at train time. However, the authors do not properly motivate the technique, the baselines are inadequate, and the paper can be sometimes unclear. For these reasons I give a score of weak reject.

---

> ### Author Response · Authors · 2021-11-19
> **Response to Reviewer H39o**
>
>
> We appreciate your constructive suggestions. The summary of your main reviews and our responses is listed below.
>
> ---
>
> **Q1.** Lack of description about why our poison algorithms work.
>
> **A1.** Thanks for pointing out that. Before we propose our method, we have discussed the necessity of robust features and a consistent label bias in **Section 4.1** and **Section 4.2**, respectively.
>
> As shown in **Section 4.3**, our poisoning algorithm could be described as an alternative procedure between two steps as follows:
>
> - generating poisons $D_{p}$ from the source model $f_{\theta_{s}}$ by
>
> $$
> \begin{aligned}
> \delta_i^p=arg\min_{\Vert \delta_i^p\Vert\leq\varepsilon_p}\ell_{\rm{CE}}(f_{\theta_s}(x_i+\delta^p_i), y^{\prime}_i), \forall x_i \in D_c.
> \end{aligned}
> $$
>
> - adversarial training of the source model $f_{\theta_s}$ such that it could robustly predict the poisoned images $(x+\delta_i^p)$ to the inducing target labels $y^\prime$, i.e.,
>
> $$
>  \min_{\theta_s} E_{(x_i+\delta_i^p,y_i)\sim D_p}\max_{\delta} \ell_{\rm{CE}}(f_{\theta_s}(x_i+\delta_i^p+\delta),y_i').
> $$
>
> In particular, ADVIN could combine the advantages of using robust features and consistent label bias for fooling adversarial training. As shown in Eq. 5, the noise is generated by the source model through a targeted PDG attack towards the targeted label $y'$ with a consistent bias. Meanwhile, the source model is adversarially trained as shown in Eq. 6. In this way, we can inject the robust features of the targeted classes $y'$ into $y$-class samples consistently. When the poisoning success rate (PSR) reaches the given threshold, the poisons are trained to have enough robust features about the target class $y'$. Therefore, when the target model is trained on the $y$-class poisoned data, it will be induced to use the $y'$-class robust features in the perturbation to predict the true label $y$. However, it will have a catastrophically poor performance on natural test data where $y$-class samples only contain $y$-class features.
>
> We have added this discussion in **Section 4.3** in the revision.
>
> ---
>
> **Q2.** Lack of discussion of the AT-pretrained method.
>
> > It would be good to start out with simple extensions of the AT-Pretrained method first.
>
> **A2.** Actually, we discussed why our poison algorithm works from the discussion of an AT-pretrained model as shown in **Section 4.1** and **Section 4.2**. As shown in **Figure 2(a)**, while an STD-pretrained model could not generate effective poisons, an AT-pretrained method succeed in decreasing the robustness of the target model to about **35\%**. Here we observe the importance of robust features from the extensive experiments. Finally, we change the AT-pretrained source model to a randomly initialized model and train it from scratch, which helps reduce the discrepancy between the source model and the target model and improve the effectiveness of our poisons.
>
> ---
>
> **Q3.** Clarity problem: Lack of discussion of Huang's work [1].
>
> **A3.**  Huang's work [1] transfers a bi-level minimization optimization problem to an alternative optimization process. Specifically, they propose a bi-level optimization problem for noise generation as the following equation:
>
> $$
> \min_{\theta_s}E_{(x_i,y_i)\sim D_c} \min_{\Vert \delta_i^p\Vert\leq\varepsilon_p}\ell_{\rm CE}\left(f_{\theta_s}(x_i+\delta^p),y_i\right),
> $$
>
> where the inner minimization seeks the $L_{p}$-norm bounded noise $\delta$ that minimizes the model’s classification loss, and the outer minimization finds the parameters $\theta$ that also minimize the model’s classification loss. To solve this, they divide it into an alternative loop of the training process. In every loop, they optimize $\delta$ over $D_c$ through PGD attack after $M$ steps of optimization of $\theta$.
>
> For clarity, we have expanded the discussion of Huang's work around the above equation in the additional **Appendix F**.
>
> [1] Hanxun Huang, Xingjun Ma, Sarah Monazam Erfani, James Bailey, Yisen Wang. Unlearnable Examples: Making Personal Data Unexploitable. In ICLR. 2021.
>
> ---
>
> **Q4.** The title of the forum page for this paper does not match the title of the pdf of this paper.
>
> **A4.** Thanks for pointing it out. We will modify the title of the forum page.
>
> ---
>
> Thanks for your review. We have added relevant discussions in the revision for better clarity following your suggestions. Please let us know if you have additional concerns.

---

> > ### Comment · Reviewer_H39o · 2021-11-28
> > **Response**
> >
> > Thank you for the response!
> > > Q2. Lack of discussion of the AT-pretrained method.
> >
> > It would be good to try the poisoning process over an ensemble to provide a strong baseline for the method. It is a much more obvious procedure than the proposed joint-training scheme and would be good as a comparison.

---

> > ### Comment · Reviewer_H39o · 2021-11-28
> > **Visualizing poisoned training images**
> >
> > After reading Reviewer eute's response I am also curious about what the poisoned images look like---it could be that one could semantically change the image with such a large perturbation ($32/255$). Can the authors post randomly sampled examples of poisoned training examples?

---

> > > ### Author Response · Authors · 2021-11-29
> > > **Further Response to Reviewer H39o**
> > >
> > > Thanks for your response! We will address your concerns as follows.
> > >
> > > ---
> > >
> > > **Q1.** Visualizing poisoned training images
> > >
> > >
> > > **A1.** We actually have shown a list of randomly selected **(not cherry-picked)** poisoned examples under different $\varepsilon$ (from $0$ to $64/255$) in **Figure 7** in Appendix D. It can be seen that $\varepsilon=32/255$ noise will not alter the belonging class because at least, humans can still tell the labels are consistent with the clean examples.
> > >
> > > In the revised paper, to provide more evidence, we provide more randomly selected poisoned examples in **Figure 8**, where we sample 10 samples for each class.  Across all 100 random samples, we find that it is still hard to find one single case that the labels are changed. This suggests that **most of (if not all) the poisoned examples** are label-consistent and semantic-preserving.
> > >
> > > ---
> > > **Q2.**  Try the poisoning process over an ensemble to provide a strong baseline for the method.
> > >
> > > **A2.** Thanks for your suggestion, and indeed this is an interesting baseline to compare. Due to the limit of time left, we cannot accomplish it now, but we are willing to include the new results in the future. We reckon that the ensemble may have little hope of winning IAT because the gap is huge between IAT and AT-pretrained (30% acc), but it could still serve as a motivating baseline to develop our iterative scheme.
> > >
> > > ---
> > > Hope this could address your concerns and we're looking forward to hearing from you!

---

> > > > ### Author Response · Authors · 2021-11-30
> > > > **Need further clarification?**
> > > >
> > > > Thanks for your constructive comments. We have tried our best to address the concerns. Is there any unclear point that we should/could further clarify? This is the last day and we are looking forward to hearing your response.

---

### Official Review · Reviewer_eute · 2021-11-02

**Correctness:** 3
**Technical Novelty And Significance:** 2
**Empirical Novelty And Significance:** 2
**Recommendation:** 5
**Confidence:** 4

**Details Of Ethics Concerns:**

There is no ethics concern.

**Main Review:**

## Strengths
1. The paper studies the failure of the existing method, for instance, how the data poison attack proposed in Huang et al. (2021) fails in the adversarial setting. This motivates a stronger type of attack that can fool adversarially learned models.

2. The paper is clearly written. The logic and content are well organized.

3. The paper presents many experiments to show the effectiveness of fooling adversarial training.

## Weaknesses
1. The findings that existing data poison attacks do not work well in adversarial training are quite trivial since those attacks focus on the normal setting but might not consider the existence of adversarial training. For instance, the error-minimization-noise in Huang et al. (2021) generates noise based on normally trained models and therefore it is very natural that it doesn't fool adversarial training. Therefore, such findings are not surprising.

2. The novelty of the proposed poison algorithm is weak. The major idea in this paper follows the error-minimization-noise in Huang et al. (2021), but it considers updating the model based on the adversarial loss defined on the adversarial samples with poison noise instead of on the clean data with noise. Although this paper proposes other techniques for improvements such as the consistent label bias, overall, the novelty is quite weak.

3. The paper doesn't discuss whether existing defense methods against data poison attacks can defend the proposed attack. Although the poison data looks visually similar to the original data, it is still possible to detect or remove such attacks. More discussion on this will be helpful.

4. The data poison effect is not that strong as shown in the main paper. If I understand correctly, in most of the experiments in the paper, the poison ratio is 100%. In practice, it is unlikely that we can control and poison the whole dataset. However, Table 8 shows that when the poison ratio is lower, the poison effect is much weaker. For instance, when 20% / 40% / 60% of the data samples are poisoned, the robust accuracy only decreases by 0% / 3% / 7%. Therefore, this suggests the proposed attack might not be very strong in practice.



**Summary Of The Paper:**

The paper proposes a type of data poison attack called adversarially inducing noise (ADVIN) that degrades the performance of adversarial training. It shows the necessity of considering the adversarial learning in the poison process and the experiments show that ADVIN can successfully make adversarial training ineffective.



**Summary Of The Review:**

The paper proposes a data poison attack to fool the adversarial training algorithms. The novelty of the proposed method is weak and the attack seems to be weak when the poison ratio is lower. I would like to recommend rejection.

---

> ### Author Response · Authors · 2021-11-19
> **Response to Reviewer eute (2/2)**
>
> **Q3.** Lack of discussion about whether defense methods against data poisoning could defend the proposed attack.
>
> **A3.** First, most existing defense algorithms against data poisoning (e.g., [1]) are mainly based on the detection of poisoned data. However, a well-known fact from [2] is that detection-based defense methods are intrinsically non-robust, as we can always bypass these detectors with the adaptive attack. Similarly, we can also design adaptive poisoning methods for those detection-based methods with error-min noise. Instead, [2] shows adversarial training is still effective at defense against the strongest attack. Moreover, recent work [3] has shown that adversarial training is indeed effective for defending against data poisoning. Thus, we directly aim at poisoning this valid defense method instead of detection-based methods. Besides, we note that our goal to devise such a poison algorithm is to prevent all the data from being exploited unauthorizedly. In this case, with 100% poison rate, detection-based methods also cannot distinguish between poisoned and clean data.
>
> [1] Nathalie Baracaldo, Bryant Chen, Heiko Ludwig, Jaehoon Amir Safavi. Mitigating Poisoning Attacks on Machine Learning Models: A Data Provenance Based Approach. In AISEC. 2017.
>
> [2] Athalye et al. Obfuscated Gradients Give a False Sense of Security:
> Circumventing Defenses to Adversarial Examples. In ICML. 2018.
>
> [3] Tao et al. Better Safe Than Sorry: Preventing Delusive Adversaries with Adversarial Training. IN NeurIPS. 2021.
>
> ---
>
> **Q4.** In Table 8, the poison effect is much weaker when the poison ratio is lower.
>
> **A4.** Importantly, we first note that the goal of our proposed poisoning method is to protect data privacy by making the training examples **unlearnable**. In comparison, previous unlearnable examples [1] are only effective against standard training, while become learnable when facing adversarial training. In comparison, our method is unlearnable under both standard and adversarial training.
>
> |Poison Percentage |0% |  |20% | |40% | | 60%| |80% | |100% | |
> |---|---|---|---|---|---|---|---|---|---|---|---|---|
> | | Natural | PGD |Natural | PGD |Natural | PGD |Natural | PGD |Natural | PGD |Natural | PGD |
> |only clean |85.14\% |51.71\% |84.05\%|49.51\%|82.61\%|46.67\%|79.03\% |42.64\% |73.98\% |35.15\% |-- |-- |
> |clean+poison |--|--|84.69\%|51.64\%|84.25\%|48.68\%| 82.57\% |44.47\% |80.57\% |37.92\%|56.52\%|0.57\% |
>
> Table 8 above actually shows that our poisoned data is still unlearnable with partial poisoning. In particular, the final performance of combining clean and poisoning data is very close to that of using clean data alone (e.g., 46.67 vs. 46.68 with 60% poisoned data). This suggests our poisoning data are indeed unlearnable under adversarial training.
>
> Therefore, our poisoning methods indeed could generate **truly learnable** examples, as we mentioned in **Section 1 & 6**.
>
> [1] Hanxun Huang, Xingjun Ma, Sarah Monazam Erfani, James Bailey, Yisen Wang. Unlearnable Examples: Making Personal Data Unexploitable. In ICLR. 2021.
>
> ---
>
> In summary, thanks for your review and hope our explanations could address your concerns. Please let us know if you have additional questions.

---

> > ### Comment · Reviewer_eute · 2021-11-25
> > **New comments**
> >
> > Thank you for the detailed responses to my initial comments. However, I still have many concerns.
> >
> > First, I would like to point out that you misunderstood my concerns about the difference between Huang et al. (2021) and the proposed ADVIN. Let me put it in this way: if we ignore the shared components such as the model training $min_{\theta}$ and error-minimization $min_{\delta^p}$, the difference between Huang et al. (2021) and the proposed ADVIN is that Huang et al. (2021) defines the loss as $L_{CE}(x_i+\delta^p, y_i)$ based on the clean sample (plus noise $\delta^p$) while ADVIN defines the loss $L_{CE}(x_i+\delta+\delta^p, y_i)$ based on the adversarial sample $x_i+\delta$ (plus noise $\delta^p$). In other words, the standard loss in Huang et al. (2021) is replaced by an adversarial loss defined on the adversarial sample. You argue that $L_{CE}(x_i+\delta^p, y_i)$ is also a variant of adversarial loss due to the existence of noise $\delta^p$, but this is not my concern because they both share this noise in order to minimize the error.
> >
> > Second, you mentioned that defense methods against poison attacks can be bypassed with the adaptive attack. It is possible, but it is not verified that the proposed method can really bypass any defense methods. Considering that the poison perturbation budge is very large (32/255), although it is claimed that the poison data are semantically similar to the original ones, it is very likely that it can be detected by defense methods. Overall, it requires more effort to show how the adaptive variants for the proposed method can bypass defense methods.
> >
> > Third, you didn't respond to my concern that when 20% / 40% / 60% of the data samples are poisoned, the robust accuracy only decreases by 0% / 3% / 7% and the clean performance only decreases by 0.5% / 1% / 3%. The poison seems to be quite weak (even with a large budge such as 32/255).

---

> > > ### Author Response · Authors · 2021-11-29
> > > **Further Response to Reviewer eute (2/2)**
> > >
> > >
> > > **Q2.** Concerns about adaptative attack and poisoning rate.
> > >
> > > **A2.** We are afraid that there are some misunderstandings of the problem setup, and we will clarify them as follows.
> > >
> > > **Problem Setup.**  We want to highlight that our method is **not like backdoor data poisoning** [1,2] that want to inject a small amount of poisoning data and lead model to the desired behavior with triggers. Instead, our poisoning method belongs to the thread of **availability attack** [3,4,5,6,7] that aims to make the poisoned data **unavailable for training**, in other words, crafting "unlearnable examples" [5]. As shown in **Section 6** as well as [5,6], this kind of poisoning could be used for preserving data privacy, i.e. "adversarial for good".
> > >
> > >
> > > **Evaluation**. It can be seen that our evaluation protocols exactly follow that of [5,6,7] and all this thread of work has not considered detection-based defense, which is interesting to explore but might be beyond the scope of this single paper. **[Edited: we have now added the adaptive attack results in the new reply!]** Besides, compared to the concurrent work [7], we additionally study the transferability of our poisons across different AT objectives (TRADES, MART) in **Table 9**, and verify the black-box effectiveness of our availability attack.
> > >
> > > **Poisoning rate results**. As our method focuses on **availability attack**, then this attack is successful as long as we can show that this part of poisoned data cannot be used for training. This is exactly the message shown in our **Table 8** (quoted above). With diffferent poisoning rate, the "clean+poison" setting achieves very similar performance to the "only clean" setting, meaning that the additional poisoned data in  "clean+poison" is of almost no use. Therefore, this result is sufficient to show the effectiveness of our poisoning method. We note that similar results are also reported in the related work, e.g., Table 2 in [5], Table 5 in [6], Table 2 in [7].
> > >
> > >
> > > To conclude, our poisoning method belongs to the thread of availability attack and we have shown its effectiveness following the standard evaluation protocols. We will elaborate more on the relevant literature to make this point more clear in the revision.
> > >
> > >
> > > Reference:
> > >
> > > [1] Li et al. Backdoor Learning. 2020.
> > >
> > > [2] Schwarzschild et al. Just How Toxic is Data Poisoning? A Unified Benchmark for Backdoor and Data Poisoning Attacks. ICML 2021.
> > >
> > > [3] Barreno et al. The security of machine learning. Machine Learning. 2010.
> > >
> > > [4] Biggio et al. Poisoning attacks against support vector machines. 2012.
> > >
> > > [5] Hanxun Huang, Xingjun Ma, Sarah Monazam Erfani, James Bailey, Yisen Wang. Unlearnable Examples: Making Personal Data Unexploitable. In ICLR. 2021.
> > >
> > > [6] Fowl et al. Adversarial Examples Make Strong Poisons. NeurIPS 2021. https://arxiv.org/pdf/2106.10807.pdf
> > >
> > > [7] Anonymous. Robust Unlearnable Examples: Protecting Data Privacy Against Adversarial Learning. https://openreview.net/forum?id=baUQQPwQiAg
> > >
> > > ---
> > > Hope our explanations above could address your concerns and we are looking forward to hearing from you.

---

> > > > ### Comment · Reviewer_eute · 2021-11-30
> > > > **Will increase my score but my concerns remain**
> > > >
> > > > Thanks for the new experiments on the evaluation of the poison detection method and the adaptive attack. This is very helpful. I would like to increase my score.
> > > >
> > > > However, my concerns about the novelty and effectiveness of the proposed method remain.
> > > >
> > > > It is claimed in your response that "With different poisoning rate, the "clean+poison" setting achieves very similar performance to the "only clean" setting, meaning that the additional poisoned data in "clean+poison" is of almost no use." However, I think this
> > > > comparison and the claim are not fully convincing.
> > > >
> > > > For instance, when the ratio is 20% in Table 8, the PGD performance on $D_c+D_p$ is 51.64% and the PGD performance on $D_c$ is 49.51%. The performance difference is about 2%. But noticing that adding the same number of clean samples as the poison samples will also only increase the performance by a few percents as showed in Table 8. Therefore, a fair comparison to conclude that the poison data is not useful is to compare when those poison data are replaced by clean samples, i.e., the performance on the whole dataset (51.71%). From this, we can conclude that when 20% of data are poisoned, the poison is not useful and the protected data can still be fully used.
> > > >
> > > > Therefore, I think the effectiveness of making the poison samples useless is marginal in these practical poison ratios such as 20%, 40%, and 60%, especially considering that the method already uses very large perturbation budgets.

---

> > > > > ### Author Response · Authors · 2021-11-30
> > > > > **Thanks and Further Response**
> > > > >
> > > > >
> > > > > We sincerely thank Reviewer eute for appreciating our added detection results. We will further address your questions about the poisoning rate as follows.
> > > > >
> > > > > As mentioned above, we carry out this comparative study following the standard practice in the related work on availability attacks [1,2,3]. Here, we provide a more detailed comparison by **quoting** their results in the following table.
> > > > >
> > > > > **Note on setup.** Because of different setups, results of different methods **should not be compared vertically**, but here we only want to highlight **the change of accuracy with different poisoning rate (PR)** for each method, i.e., **horizontally**.
> > > > >
> > > > > | Method | PR=0% |  | PR=20% | |
> > > > > | ---| ---| ---| ---| ---|
> > > > > | - | only clean | clean+poison | only clean | clean+poison |
> > > > > | Huang et al [1] | 94.95 | 94.95 | 93.75 | 94.38 |
> > > > > | Fowl et al [2] | 86.00 |86.00 | 82.30 | 85.4 |
> > > > > | Anonymous [3] | 89.51 |  89.51 | 88.17 | 89.6 |
> > > > > | ADVIN (ours) | 85.14 | 85.14 | 84.05 | 84.69 |
> > > > >
> > > > > **Analysis.** We can see that, when the poisoning rate is low (20%), all availability attacks will be less effective as their "clean+poison" will be a little higher than "only clean", and some even outperforms the baseline (89.6 v.s. 89.51) [3]. In our ADVIN, as you have noticed, there is also a slight increase in accuracy, but it is still lower than the full clean baseline (84.69 v.s. 85.14).
> > > > >
> > > > > Therefore, we could say that existing availability attacks (including ours) are less effective under a low poisoning rate, and these methods are more effective with a large rate, which is also the main focus of our paper.
> > > > >
> > > > >
> > > > > Thanks again for your kind reply and hope you could take this into consideration.
> > > > >
> > > > >
> > > > > Reference:
> > > > >
> > > > > [1] Hanxun Huang, Xingjun Ma, Sarah Monazam Erfani, James Bailey, Yisen Wang. Unlearnable Examples: Making Personal Data Unexploitable. In ICLR. 2021.
> > > > >
> > > > > [2] Fowl et al. Adversarial Examples Make Strong Poisons. NeurIPS 2021. https://arxiv.org/pdf/2106.10807.pdf
> > > > >
> > > > > [3] Anonymous. Robust Unlearnable Examples: Protecting Data Privacy Against Adversarial Learning. https://openreview.net/forum?id=baUQQPwQiAg

---

> > > ### Author Response · Authors · 2021-11-29
> > > **Further Response to Reviewer eute (1/2)**
> > >
> > >
> > > Thanks for appreciating our response. We will address your further concerns as follows.
> > >
> > >
> > > ---
> > > **Q1.** You misunderstood my concerns about the difference between Huang et al. (2021) and the proposed ADVIN.
> > >
> > > **A1.** Thanks for your clarification and we now get your point. Indeed, we can tell the two methods (Error-min [1] and our ADVIN) apart by noting the difference between **inner-loop adversary $\delta$** and **outer-loop adversary $\delta_p$**, in the following ADVIN objective,
> > > $$\text{ADVIN (ours):~~~}\min_{\delta_p}\min_{\theta}E_{x,y}\max_{\delta}\ell_{CE}(x+\delta_p+\delta,y').$$
> > > But we still want to highlight that this objective is still quite different from the min-min objective of [1]:
> > > $$\text{min-min [1]:~~~}:\min_{\delta_p}\min_{\theta}E_{x,y}\ell_{CE}(x+\delta_p,y).$$
> > > - **Different targets $y'$ vs. $y$**: while error-min studies minimization over the original label $y$, we investigate adding **inducing labels** $y'$ in Section 4.2 and Table 1 shows that it is much more effective than error-min.
> > > - **Inner-loop adversary**: in Section 4.1, we show that an inner-loop adversary is necessary to be resistant to adversarial training, while error-min only considers outer-loop adversary.
> > >
> > > We also note that there is a concurrent work that also aims for crafting poisons against AT to protect data privacy, which is instead a direct extension of Huang et al's method [2] to a robust min-min loss. In comparison, we devise a new label inducing objective (ADVIN) that could induce AT to much lower robustness (0.53%) under a normal AT setting with $\varepsilon=8/255$, while theirs only considers weak adversaries ($\varepsilon=4/255$ at most).
> > >
> > > Therefore, we believe that this line of research highlights the necessity to study AT poisoning and we believe that our method also contributes to new effective techniques. Please take this into your consideration.
> > >
> > > Reference:
> > >
> > > [1] Hanxun Huang, Xingjun Ma, Sarah Monazam Erfani, James Bailey, Yisen Wang. Unlearnable Examples: Making Personal Data Unexploitable. In ICLR. 2021.
> > >
> > > [2] Anonymous. Robust Unlearnable Examples: Protecting Data Privacy Against Adversarial Learning. https://openreview.net/forum?id=baUQQPwQiAg

---

> > > ### Author Response · Authors · 2021-11-29
> > > **Added adaptive attack results**
> > >
> > >
> > > Following your suggestions, we further conduct experiments to study whether detection-based methods can be bypassed by adaptive poisoning attack with our ADVIN.
> > >
> > > **Setup.** Here we consider the widely used MMD-based detection method [1] for adversarial examples that is also considered in CW's adaptive attack paper [2]. For adaptive attack, we mix ADVIN loss and MMD loss with 1:1 to train the source model, and generated poisoned data with the source model. We then evaluate the MMD score and p-value using the detection script provided by CW [3]. A small p-value (lower than 0.05) indicates that the data are detected as adversarial data.
> > >
> > > **Results.** We compare clean data, ADVIN (non-adaptive) and ADVIN (adaptive) in the following table.
> > >
> > > | Data | MMD | p-value | Detected to be Adversarial |
> > > | ---| ---| ---| --- |
> > > | Clean | 4.76e-4 | 0.316 | No|
> > > |ADVIN (non-adaptive) | 3.72e-3 | 0.0474 | Yes |
> > > |**ADVIN (adaptive)** | **3.61e-4** |**0.337**| **No** |
> > >
> > > From the table, we can see that the original ADVIN (non-adaptive) can indeed be detected as adversarial examples as it has a large MMD. Instead, by incorporating MMD into our ADVIN loss, adaptively poisoned data can bypass the detection method with a much smaller MMD value. Meanwhile, the generated poisons still effectively degrade the AT robustness by a large margin, from 51.7% to 24.1%.
> > >
> > > Therefore, our ADVIN poisoning method is indeed effective against detection-based defense with adaptive poisoning, and thereby we mainly consider AT-based defense in our evaluation. We will add this result to our paper for completeness.
> > >
> > > Hope this could address your concerns and we are looking forward to hearing from you!
> > >
> > > Reference:
> > >
> > > [1] Grosse et al. On the (Statistical) Detection of Adversarial Examples. 2017.
> > >
> > > [2] Carlini and Wagner.  Adversarial Examples Are Not Easily Detected: Bypassing Ten Detection Methods. AISec'17.
> > >
> > > [3] MMD detection code. https://github.com/carlini/nn_breaking_detection/blob/master/maximum_mean_discrepancy.py

---

> ### Author Response · Authors · 2021-11-19
> **Response to Reviewer eute (1/2)**
>
> Thanks for appreciating the clarify and solidness of our review, though there might be some understanding of our method and related work. We address your main concerns as follows.
>
> ---
>
> **Q1.** Lack of novelty compared to Huang et al. [1].
>
> > Existing methods only focus on normal settings and generate noise based on normally trained, and thus it is not surprising they do not work in adversarial training.
>
> **A1.** We are afraid there might be some misunderstandings of our method and Huang et al.
>
> **First**, Huang et al's method **also relies on (a variant of) adversarial training** instead of normally trained models. In particular, reviewing Huang et al's error-minimizing process,
>
> $$ \min_\theta E_{(x_i,y_i)\sim D_c} \min_{\delta^p} \ell_{\rm CE}\left(f_{\theta_s}(x_i+\delta^p),y_i\right), $$
>
> which also involves adversarial samples, while the difference to AT is the min-min instead of the min-max objective. In either case, it is far from standard training models.
>
> Therefore, the key difference between our proposed ADVIN and [1] is **not about standard training v.s. adversarial training**, but that we develop a different adversarial training method for injecting **inducing noise from a consistent class**. Our method not only explains the effectiveness of error-min [1], but also improves over theirs by a large margin (13.4% $\to$ 0.57%).
>
> **Second**, even though focusing on the vanilla adversarial training, existing method, e.g. [2], can also be applied to poison ST while failing for fooling AT, as discussed in **Section 4.1 & 4.3**. For example, Figure 3 shows that poisoned generated with an AT-pretrained model (as in [2]) can not effectively fool AT: it only leads to a slight drop of robust accuracy comparable to ours (**35% v.s. 1%**). Therefore, using robust features as poisons is critical but insufficient for noise generation.
>
> Therefore, our ADVIN is a non-trivial solution to poisoning AT with a new poisoning adversarial objective (Eq. 6 & 7) and we are the first to fool AT to be as low as 0.53% accuracy.
>
> [1] Hanxun Huang, Xingjun Ma, Sarah Monazam Erfani, James Bailey, Yisen Wang. Unlearnable Examples: Making Personal Data Unexploitable. In ICLR. 2021.
>
> [2] Liam Fowl, Micah Goldblum, Ping-yeh Chiang, Jonas Geiping, Wojtek Czaja, Tom Goldstein. Adversarial examples make strong poisons. In NeurIPS. 2021
>
>
> ---
>
> **Q2.** The Novelty of our proposed poison algorithm.
>
> **A2.** We highlight the following novel contributions of our proposed ADVIN.
>
> **First**, we aim to study and fulfil the goal of fooling adversarial training. In comparison, prior work like [1] can only be applied to poisoning standard training and can be easily defended by adversarial training. In particular, we have given the analysis about reasons why Error-min noise [1] fails to make adversarial training ineffective in **Section 3**.
>
>
> **Second**, our algorithm has an entirely different working mechanism. While Huang's work begins with forcing the loss during learning to decrease to near zero, our work utilizes the robust features of another class to fool the model to learn a wrong mapping between poisons and the clean labels.
>
> 1) As discussed in Section 4.1, robust features, coming from an adversarially trained source model, ensure that the poisons would not be removed by the perturbations of adversarial training for the target model.
> 2) As discussed in Section 4.2, a consistent label bias forces the source model to extract the robust features of another class (different from the original label) as poisons.
>
> Although some similarities exist between our work and Huang's [1], the crucial roles of a target label consistency and an adversarial induction process are our key contributions and we have shown they are critical for crafting an effective poisoning.
>
> [1] Hanxun Huang, Xingjun Ma, Sarah Monazam Erfani, James Bailey, Yisen Wang. Unlearnable Examples: Making Personal Data Unexploitable. In ICLR. 2021.

---

### Decision · Program_Chairs · 2022-01-20

**Decision:**

Reject

**Comment:**

The paper shows that adversarial training can be fooled to have robust test accuracy < 1% with a new type of poisoning attack ADVIN on the CIFAR-10 dataset, even though the robust training accuracy > 90%. This requires 100% poisoning rate, though the claim is that the poisoned data is 'semantically similar' to the original data. This is an interesting research direction. Questions were raised about novelty as well as whether these poisoned data could be detected. During the rebuttal phase the authors provide some evidence that with adaptive attacks detection could be evaded (as expected). The authors are encouraged to take all comments into account and update the paper as indicated in the rebuttal for any future revision.